# Gel Fuels: Preparing, Rheology, Atomization, Combustion

**Dmitrii Glushkov \***, **Kristina Paushkina** and **Andrei Pleshko**

Heat Mass Transfer Laboratory, National Research Tomsk Polytechnic University, Tomsk 634050, Russia
\* Correspondence: dmitriyog@tpu.ru; Tel.: +7-(3822)701-777 (ext. 1953)

**Abstract:** The review presents the results of experimental and theoretical studies obtained in recent years within the framework of the main areas of research of gel fuels: choice of component composition and substantiation of fuel preparation technologies and their rheological characteristics; fuel transportation and atomization processes; and the processes of their ignition and combustion. The main advantages of gel fuels in comparison with widely used liquid and solid fuels are considered. The advantages and disadvantages of known experimental approaches to the study of the combustion processes of gel fuels are analyzed. The well-known physical and mathematical models of gel fuels ignition are given, including those under conditions of melt droplets dispersion. The tendencies of further development of gel fuels within the framework of the combustion theory are formulated.

**Keywords:** gel fuel; dispersion; combustion; rheology; atomization

## 1. Introduction

At present, three main types of fuels are widely used in practice: liquid [1,2], solid [3,4] and gaseous [5,6]. Liquid fuels (gasoline, diesel fuel, kerosene, hydrazine, heptyl, alcohols, naphtha, benzene-gasoline mixtures, fuel oil, kerosene and methoxydiethylborane/tetrahydrofuran and others), as a rule, are used in internal combustion engines (motor transport, shipbuilding) [7], jet engines (aviation) [8], liquid-propellant rocket engines [9] and technological installations in thermal power engineering [10]. Solid fuels (coals, combustible shale, metallized composite solid fuels and others) are used as energy resources in coal-fired steam and water-heating boilers [11], blast furnaces [12] and solid rocket engines [13]. Gaseous fuels (methane, propane, dimethyl ether, coke oven gas and others) are widely used in various applications, but the most efficient energy potential of these fuels is used in gas turbines [14].

Along with the aforementioned fuels, in recent years, in transport technology and in the energy sector, a promising direction of development is the design of effective compositions and methods for obtaining various types of composite fuels: synthesis gas [15], coal-water slurries [16], coal-water slurries with petrochemicals [17], suspension fuels [18], gel fuels [19,20] and fuel briquettes [21]. This is mainly attributable to the tasks of improving the environmental, energy, economic and operational characteristics of both technological systems of devices in transport and energy, and the processes of storage, transportation and combustion of new fuels types.

In recent years, the prospects for space exploration have been the main incentives for developments in the rocket and space industry. The solution for this problem requires an integrated approach to conducting fundamental research and using their results in practice to make rational, technical decisions when conducting development work. One of the main problems is the need to develop new fuels, study their properties, as well as study the physicochemical processes that occur during their ignition and combustion.

Solid and liquid propellants were widely used in rocket and space technology at the end of the XX and the beginning of the XXI century. Each of them has its own advantages and disadvantages compared to one other. To date, the potential of such fuels is almost completely exhausted, according to many experts [22–24]. Gel fuels can become

an alternative energy resource within the framework of the implementation of the program for developments in rocket and space technology. These propellants combine the advantages of solid and liquid rocket propellants [24], of which the main ones are related to safety aspects (a small amount of vapor is released during leaks compared to liquid fuels; indifference to shock, friction and electrostatic discharge; minimal risk of accidental ignition, because the combustion process is sufficiently well controlled and can be stopped; the elastic-deformable or viscous-plastic state prevents the formation of cracks in the fuel charge, so their development does not affect the increase in the combustion area, uncontrolled combustion or explosion), energy efficiency (relatively high specific thrust impulse compared to composite solid propellants) and storage (long-term storage without maintaining special conditions; flexible packaging; relatively low proportion of solid fine particle settling in high-viscosity formulations compared to liquid fuels). Gel fuels have higher energy characteristics than composite solid propellants; in particular, a specific thrust impulse is about 3000–3500 m/s [25]. Replacing a composite solid propellant with a gel fuel will make it relatively easy to implement dynamic control of engine thrust during the time in practice [24]. Furthermore, gel fuels, compared to liquid fuels, have lower fire hazard rates because of the minimization of evaporation losses and leakage during storage. Gel fuel components and their combustion products tend to have a lower environmental impact than typical liquid propellants [26].

The aggregate state of gel fuel can differ significantly depending on its purpose and environmental characteristics—from liquid (with high viscosity) to solid (elastically deformable). Within the framework of ignition theories of solid and liquid condensed substances [27–31], the properties of composite solid propellants and liquid fuels have been studied in full, and mathematical models have been developed to predict the characteristics of physicochemical transformations that occur when they are heated. The results of experimental studies [22,23] indicate a fairly significant difference in the regularities and characteristics of the physicochemical processes occurring during the ignition and combustion of gel fuels, compared with typical solid and liquid fuels. Therefore, an urgent task is a comprehensive review of studies on gel fuels. An analysis of publications on this topic in international periodicals made it possible to identify the following main areas of research conducted to date:

1.　Preparation of fuel compositions based on various components [32–37];
2.　Study of the rheological characteristics of fuels [35,36,38–41];
3.　Study of the processes of gel fuels atomization [41–44];
4.　Study of the processes of fuel composition, ignition and combustion [22,23,45–48];
5.　Application of fuels in practice [32,49–53].

Below are the results of the analysis of the main achievements in recent years for each of the abovementioned research areas.

This review is focused on a brief coverage of modern achievements in the field of gel fuels. Only the most important results of leading experts in the field of gel fuels are presented here, with an emphasis on the published research results of recent years, including the results of studies that have not yet been published. Within the framework of the topic of gel fuels, this review is primarily focused on the study of elastically deformable fuel compositions, whereas most other studies are devoted to plastically deformable gel fuels.

## 2. Fuel Preparation

Physically, gel fuels are rheologically modified liquids, the properties of which are changed by adding various thickeners (gelling agents) [24,54–56]. Depending on the type of thickeners, it is currently conventionally accepted [24,54,55] in the scientific periodical literature as the distinguishing of two fundamentally different (in terms of rheological properties and physical and mechanical characteristics) types of gel fuels. On the one hand, these are "soft substances" (Figure 1a) that exhibit the physical and mechanical characteristics of both a solid and a liquid, depending on the level of shear stress [24,54].

On the other hand, these are materials with a three-dimensional elastically deformable solid framework (Figure 1b) containing finely dispersed liquid drops in the matrix cells [55].

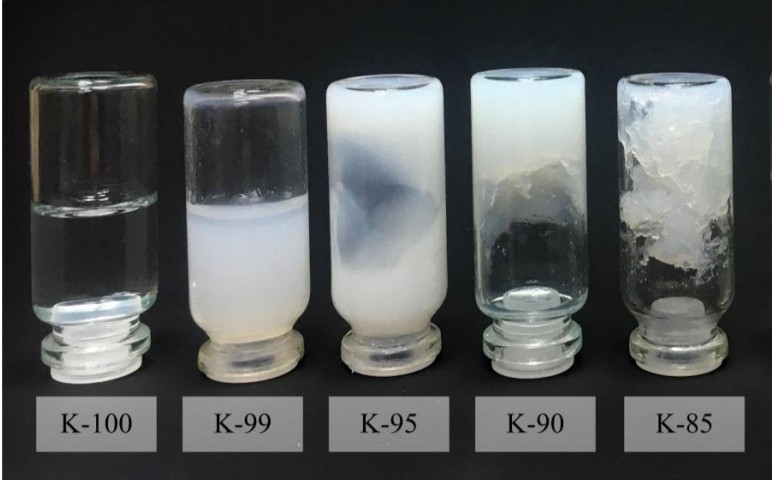

(**a**)

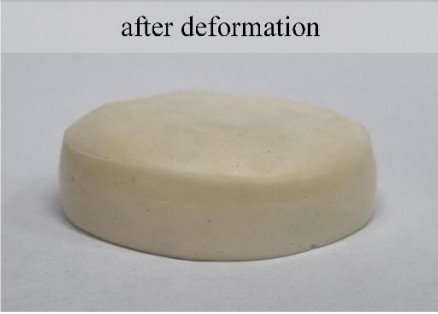

(**b**)

**Figure 1.** Appearance of gel fuels: (**a**) Liquid and plastically deformable fuel compositions based on kerosene: K-100—100 % kerosene, K-99—99 wt% kerosene, 1 wt% silicon dioxide, K-95—95 wt% kerosene, 5 wt% silicon dioxide, K-90—90 wt% kerosene, 10 wt% silicon dioxide, K-85—85 wt% kerosene, 15 wt% silicon dioxide; (**b**) Elastically deformable: 50 vol% oil + 48 vol% aqueous solution of PVA (10 wt%) + 2 vol% emulsifier.

The first type of fuel is obtained, as a rule, by adding organic and inorganic thickeners to combustible liquids, and the second is obtained by adding polymeric thickeners. Rheological properties, physico-mechanical and physico-chemical characteristics of combustible liquids change quite significantly after thickening. After obtaining the gel fuels and for a certain period of time during storage, their viscosity increases by several times compared to combustible liquids in the initial state, and the density increases significantly, especially if finely dispersed metal particles are added to the fuel composition [56]. These components also contribute to the growth of the thermal effect of the combustion process [57] and the specific energy density [24]. Thus, the production and practical application of gel fuels can provide a high level of control over operational characteristics, uniformity and reproducibility of fuel compositions, as well as energy characteristics.

Mainly potential technologies for the use of gel fuels, for example, in rocket and ramjet engines, afterburners, involve the implementation of the processes of their transportation through pipelines, supply to the combustion chamber and organization of direct combustion, similar to the processes characteristic of typical liquid fuels in the initial state [24,54–57]. It is worth noting that in addition to thickening flammable liquids, gel oxidizers are also used in practice to solve some problems [58,59]. In the early stages of development of gel fuels in the middle of the XX century, they were considered solely as a

replacement for mixed solid fuels [58–63]. Accordingly, the corresponding formulations of the first gel fuels and oxidizers were developed [58–63].

However, later, it was proposed to use gel propellants instead of liquid propellants [54,64], since the thickening of the latter made it possible to minimize their main operational disadvantages, as well as to improve the energy characteristics mainly by adding metal powders, the particles of which can be fairly evenly distributed in the volume of the thickened combustible liquid. This has led to gel fuels being predominantly characterized as non-Newtonian fluids. The compositions most widely used in experimental studies are given in references Table 1.

**Table 1.** Compositions of typical gel fuels.

| Rocket Fuel | Thickener [a] | Additives [a,b,c] | Notes | Ref. |
|---|---|---|---|---|
| Water (98.5%) | Carbopol 941 (0.5%) (2–7 μm) | NaOH (1%) | Mechanical mixing | [23] |
| JP-5 JP-8 IRFNA HP | AO (2–5%) $SiO_2$ (4–8%) $SiO_2$ (3–5%) $SiO_2$ (5–7%) | – | Mechanical mixing | [65] |
| Jet A1 (85%) | Thixatrol 289 (7.5%) | MIAK (7.5%) | Mechanical mixing | [66] |
| Jet A1 (NA) | Thixcin R (4–7%) | NA | Mixing at 40–60 °C for 2.5 h at 1000 rpm. | [67] |
| MMH | HPC (3%) Aerosil (6%) Hybrid Gel (2–3%) | – | Mechanical/acoustic mixing at 60 °C | [68] |
| Jet A1 (85%) | Thixatrol ST (7.5%) (23 μm) | Xylene (7.5%) | Mixing at 63 °C for 1 h at 1200 rpm. | [34] |
| ISROsene (85–65%) | Thixatrol ST (7.5%) (23 μm) | Xylene (7.5%) Nano aluminum oxide (50–150 nm) Oleic acid (10%) | Mixing at 63 °C for 1 h at 1200 rpm. | [69,70] |
| RP-1 | Carbosil (5%) (44 μm) | 0–55% Alex (100 nm) Tween 85 (0–1.3%) | – | [71] |

Notes. [a]—relative ratios of components in %wt. [b]—weight percent surfactant dependent on metal content. [c]—liquid additives are used as solvents for thickeners.

As a rule, the main component of these fuel compositions are flammable liquids used as fuels in aerospace technology, such as liquefied hydrogen [72–74]; hypergolic fuels, including hydrazine [36,75–77], monomethylhydrazine (MMH) [36,63,64,75,77–79], unsymmetrical dimethylhydrazine (UDMH) [22,36,63], tetramethylethylenediamine (TMEDA) [80]; various grades of kerosene-based jet fuel JP (JP-1, JP-5, JP-8 and JP-10) [63,75–77] and Jet A1 [23,34,50,81,82], RP-1 [54,71]; and alcohol-containing fuels based on methanol [58] and ethanol [58,83,84]. Other common liquid components of gel fuels are isopropanol [58], oils of petroleum origin [61], spent organic solvents (benzene, acetone and chlorinated solvents) [61], nitromethane [85], ethanolamine [86], aniline [63], nitropropane [63], isooctane [87], hexane [73,74], propane [74], ethane [74], paraffin [50], glycerol [88] and ammonium ethyl acetate [89].

Oxidizers used in gel form are mainly used in conjunction with hypergolic fuels. Among such oxidants, red fuming nitric acid (RFNA) [62,87] inhibited red fuming nitric acid (IRFNA) [62,64,75,77] and hydrogen peroxide [75–77,87,89], as well as cryogenic substances such as substituted bromine trifluoride [62], bromine pentafluoride-chlorine trifluoride [62], liquefied oxygen [89] and oxygen difluoride; a large number of nitroalkanes and nitrogen tetroxide [59] can be noted as well.

A significant amount of research related to the development of fuel formulations is aimed at identifying suitable types of thickeners for specific combustible liquids and oxidizers [24,77]. As a rule, most thickeners are commercially available manufactured compounds. Both inorganic [90] and organic thickeners are used, many of which are polymeric [91]. For thickeners, typical selection criteria apply [90–92]: they must be compounds that are applicable at low concentrations; they must be combustible, provide a reversible system; and thickeners should be non-aggressive, environmentally friendly and economically affordable.

Inorganic thickeners, such as silica, work well with fuels and oxidizers [91]. However, they are inert and, therefore, their use obviously leads to losses and a decrease in the energy efficiency of the fuel during combustion. In addition, their tendency to form covalent bonds reduces the reversibility of the gel fuel and negatively affects its flow characteristics.

Polymeric thickeners such as Carbopol, cellulose compounds, Xanthan and others are used as aqueous solutions to thicken combustible liquids. These compounds are straight or branched chain polymers and form a strong entangled or cross-linked network through hydrogen bonds when dissolved in water [91]. Cellulosic compounds and agar-agar can only thicken fuels. Although the rheology of these polymer gels exhibits shear thinning characteristics, the molecular bonding of the polymer chains to water prevents liquid breakdown and subsequent aerosol formation, which impairs atomization. However, atomization can be achieved for fuels containing agar-agar as a thickener, which absorbs water and forms a network around the liquid rather than binding to its molecules [92]. These examples illustrate the importance of the thickener–liquid interaction in fuel atomization and highlight the need for proper thickener selection.

Unlike polymer gels, low molecular weight thickeners are used to obtain some organic gels based on organic liquids [93]. Organogels resemble polymer gels in the structure of molecular bonds that form a matrix, in the cells of which finely dispersed liquid drops are located. They are easier to spray compared to polymer gels because of a component with reduced tensile viscosity. Moreover, they burn together with liquid fuel and oxidizers. For example, the castor oil derivative Thixcin R is used to thicken a suspension of coal and oil [61], as well as Jet A1 [94]. Other castor oil derivatives are widely used as a non-polymeric organic thickener—Thixatrol [23,34,50,69,70,82,94], Thixatrol ST [23,34,50,82,91,95], MPA 60 [96], Thixatrol 289 [97,98], Thixcin R [61,94], Thixatrol Plus [99].

Among inorganic thickeners, and most likely among all types of thickeners used for the preparation of gel fuels, various forms of silicon dioxide are most widely used [72,75,79,86,87,99]; for example Cabosil and Aerosil. It is worth noting that the results of numerous studies indicate the need for caution when choosing these commercially available silica gel thickeners for the preparation of gel fuels, since the latter do not always satisfy the basic typical requirements for rheological properties and physical and mechanical characteristics. To date, a large number of combinations have been proposed between widely used liquid fuels and thickeners, but only a small part of these fuel compositions has ever been used in experimental studies of the gel fuels spraying and combustion.

Equally important components of gel fuels are energy additives, for example, metal powders. The concentration of the metal is the determining factor if it is used as an energy additive. The typical range of metal content in fuel by mass is 5–60% [91,98,100]. The justification for the advantage of metallized fuel compositions is to obtain suspensions that have a greater value of thrust per kilogram of oxidizer (or air). Preference is given to metals with the highest volume and mass values of combustion heat [101–104]. It is known that Mg is characterized by the highest thrust per kilogram of air at stoichiometric mixture ratios, followed by Al [101]. In practice, $Al_2O_3$ is sticky and melts at the flame temperature of a typical gel fuel [100,101], whereas boron combustion is difficult to initiate [102]. Magnesium oxides do not melt and, therefore, Mg is a promising component for metallized gel fuels. The efficient use of boron requires the organization of a more complex fuel combustion mode [103,104].

The components range and gel fuels compositions are diverse, so the method of their preparation in each specific case can be quite different, especially when using thickeners of various origins (polymeric, organic, and inorganic). However, most techniques for preparing gel fuels rely on the mixing of thickener, solvent (optional), metal particles, catalysts, stabilizing surfactants (if metals are added) and liquid fuel at room conditions or elevated temperatures using mechanical or acoustic methods. The order of adding the listed components to the mixture, their concentrations and mixing times are quite significantly different in each specific case [34,75,94,105–108]. The main stages and conditions for the preparation of gel fuels are four possible combinations of combustible liquid component/oxidizers and thickeners: fuel–organic (polymeric) thickener, oxidizer–inorganic thickener, fuel–inorganic thickener and fuel–organic (non-polymeric) dispersed thickener. It should be noted that compositions based on organic thickeners and oxidizing agents are of no practical interest, since the vast majority of oxidizing agents are highly reactive with organic thickeners [105].

In [19,109,110], methods for the preparation of gel fuels are presented by thickening waste oils of petroleum origin with a polymeric thickener (without additional components and with the addition of finely dispersed solid combustible particles), as well as by thickening kerosene with an inorganic thickener (silicon dioxide).

Based on the analysis of references [19,34,75,94,105–110], it was concluded that the component composition of gel fuels is very diverse and, in each specific case, is determined by the availability of components, such as a thickener (organic, inorganic or polymeric) and the necessary rheological properties or physical and mechanical characteristics of the final product, as a rule. The latter, in turn, have a direct impact on all subsequent stages of the gel fuel life cycle, in particular, the processes of its storage, transportation through fuel lines, spraying in combustion chambers and direct combustion.

## 3. Rheology

Gel fuels are characterized by non-Newtonian properties. According to their rheological properties, they can be conditionally divided into viscoelastic and viscoplastic substances [111]. After preparation, the gel fuel is viscoelastic, but under conditions of exceeding the limiting stress (yield stress), it exhibits viscoplastic properties. Using the approaches and methods of rheology and physical chemistry, the deformation under the action of stress is studied, including flow and elastic deformation. Structural and mechanical properties of gel fuels are studied within the framework of colloid chemistry models using research methods adopted in rheology. The rheological characteristics given in this paper are relevant for plastically deformable fuels and for elastically deformable ones in a molten (liquefied) state suitable for transportation and atomization.

### 3.1. Shear Strain

The shear of gel fuel occurs at all stages of the technological process of its use (from preparation to combustion). Under shear conditions, the geometric shape of a unit of mass changes at a constant volume.

A typical process of burning gel fuel (Figure 2) in practice consists of six main stages [45,112–115], which successively replace each other: (1) storage; (2) moving from the fuel tank to the transport channel; (3) movement along the fuel pipeline (mainly through pipelines); (4) atomization (or dispersion); (5) movement of fuel fragments (particles or drops) in the combustion chamber; (6) melting, evaporation and combustion of fuel. Storage is the longest period during which the gel fuel causes the least intake due to gravity.

At the second stage, under the conditions of moving gel fuel from the tank to the transport channel, the shear rate is $2.4\,\text{s}^{-1}$ at a fuel flow rate of $0.015\,\text{m/s}$ and a characteristic time of about 53 s. Stages three to six are fast-paced, and each is completed in less than 1 s. The shear rate at each of these stages varies over a wide range and depends on the design of technological elements and the speed of fuel movement in them.

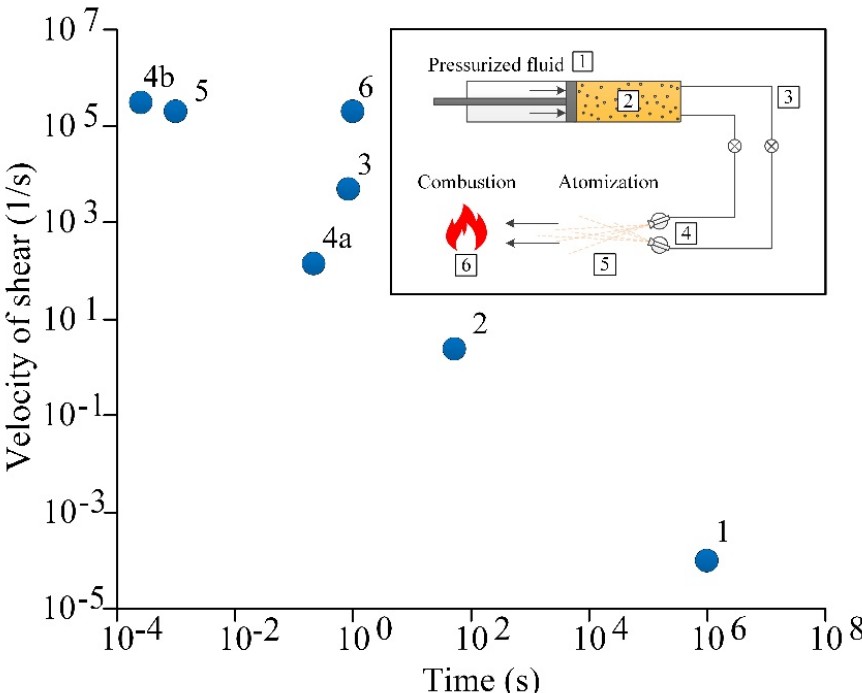

**Figure 2.** Shear rate and characteristic time of the stages implementation of the typical gel fuel life cycle [45,112–115]: 1—storage; 2—moving from the fuel tank to the transport channel; 3—moving in fuel line; 4—atomization with nozzle (4a) or hydraulic nozzle (4b); 5—movement of fuel fragments (particles or droplets) in the combustion chamber; 6—melting, evaporation and combustion of fuel.

At the third stage, the shear rate recorded under the conditions of the movement of gel fuel through the pipeline at a speed of 2.4 m/s for 0.84 s is $4.8 \times 10^3$ s$^{-1}$.

When dispersing fuel at the fourth stage, the speed of movement of gel fuel fragments depends on the method of atomization and the design of the corresponding spray device. It is known [45,112–115] that when organizing atomization with the help of nozzles, the speed is about 0.23 m/s and the time of movement is 0.23 s. Under such conditions, the shear rate is close to 141 s$^{-1}$ (Figure 2, point 4a). When organizing atomization by means of a hydraulic nozzle (Figure 2, point 4b), the shear rate reaches $3 \times 10^5$ s$^{-1}$ at a fuel particle velocity of 38.2 m/s and a time of $2.6 \times 10^{-4}$ s.

At the fifth and sixth stages, the velocities of the fuel particles are, as a rule, the same. It was registered [45,112–115] that under the conditions of evaporation and combustion of fuel particles when they move through the combustion chamber at a speed of 20 m/s for 1 s, the shear rate is $2 \times 10^5$ s$^{-1}$.

The shear rate of the gel fuel reaches its maximum values in the combustion chamber because the viscosity of the fuel decreases significantly with increasing temperature. When particles of gel fuel move in a stream of high-temperature gases, a combination of the following interrelated physical and chemical processes occurs: (1) fuel melting; (2) free surface evaporation; (3) formation of a gas–vapor mixture in the vicinity of a melt droplet (it should be noted that the temperature of the vapor entering the oxidizer medium does not exceed the boiling temperature of the combustible liquid, on the basis of which the gel fuel is prepared); (4) in the presence of solid inclusions in the composition of the gel fuel, relatively cold metal particles enter, together with vapors, into the oxidizer medium; (5) as a result of the heating of combustible vapors and solid combustible components, they ignite and burn out. A more detailed description of the combustion stages of gels will be given in Section 5.

The shear rate at each of these stages varies over a wide range and depends on the design of the technological elements and the speed of fuel movement in them. Shear rate is an important characteristic of gel fuel, which determines not only its shape, but also

rheological properties throughout the life cycle of existence from preparation to combustion [45,112–115].

*3.2. Rheological and Physical–Mechanical Characteristics*

Gel fuels are rheologically complex non-Newtonian fluids. Viscosity, yield strength, thixotropy and viscoelasticity are the most important rheological characteristics that determine not only the shape of the gel fuel, but also the stability of properties at all stages of its life cycle. During this period, it is especially important to control the viscosity of the gel fuel. The yield stress is important to control at the stage of storage, spraying and combustion. Control of thixotropy parameters is important at the stage of preparation and storage, and viscoelasticity at the stage of combustion.

Structural-mechanical properties of gel fuels can be estimated by constructing rheological curves (flow and viscosity curves). The construction of rheological curves of gel fuels is carried out according to the results of measurements obtained using a viscometer or rheometer. The flow equations of Herschel-Bulkley [116], Casson [117], Bingham [118], Ostwald [119], Carro-Yashida [120,121], the modified Herschel-Bulkley equation [65,122,123], three-parameter equation [124] and the modified Ostwald equation [42] are most often used to describe the rheological behavior of gel fuels. The wide possibilities of varying the composition of gel fuels do not allow for deriving a universal expression for describing the rheological and physical–mechanical characteristics. Experimental points of fuels with similar compositions can be approximated by different equations of flow or viscosity. The choice of equation depends on the researcher.

3.2.1. Viscosity

Shear viscosity (or dynamic viscosity, effective viscosity, apparent viscosity) is a physical quantity that describes the resistance of a substance to a viscous shear flow or energy loss in a viscous shear flow, and characterizes the internal friction of a gel fuel [111]. This friction occurs between the layers of fuel as a result of its movement. With an increase in shear rate, the viscosity of gel fuels decreases, tending to a constant value corresponding to the complete liquefaction (degradation) of the fuel.

During their life cycle, gel fuels are subject to different shear rates. During storage, when the fuel is exposed to low-intensity vibration, the shear rate does not exceed $10^{-2}$ s$^{-1}$ [45,112–115]. The shear rate reaches up to $10^3$ s$^{-1}$ [45,112–115] during loading/unloading, moving gel fuel from the tank to the transport channel, as well as when it moves along the fuel line. When a significant momentum is transferred to the gelled fuel under dispersing conditions, for example, through a hydraulic nozzle, the shear rate exceeds $10^3$ s$^{-1}$. The shear rate exceeds $10^5$ s$^{-1}$ [45,112–115] under conditions of movement of fine fuel particles in the volume of the combustion chamber, as well as during the evaporation and combustion of fuel for a relatively short period of time. Conventionally, three ranges of shear rate changes can be distinguished for typical gel fuels corresponding to different stages of their life cycle:

1. Low (less than 1 s$^{-1}$);
2. Medium (from 1 to $10^5$ s$^{-1}$);
3. High (more than $10^5$ s$^{-1}$).

It is impossible to establish a strict dependence on the viscosity of the gel fuel on the shear rate, since the viscosity, in addition to the shear rate, is significantly affected by the component composition [125]. Experimental results [69,70,113] are known on the study of the change in viscosity (determined by pressure drop) on the shear rate of xylene gel (Jet A1 jet fuel, Thixatrol ST organic thickener) [113] and gel propellant with aluminum oxide particles (ISROsene kerosene, Thixatrol ST organic thickener, xylene) [69,70] in the conditions of movement along the fuel line to the spray device and during their atomization. Detailed information on methods and means of measuring tensile viscosity is presented in the reference [126].

Experimental results [69,70,113] are known on the study of the change in viscosity (determined from the pressure drop) on the shear rate of xylene gel (Jet A1 liquid jet fuel, Thixatrol ST organic thickener) [113] and gel propellant with aluminum oxide particles (ISROsene kerosene, Thixatrol ST organic thickener, xylene) [69,70] while moving along the fuel line to the spray device and during their atomization (Figure 3).

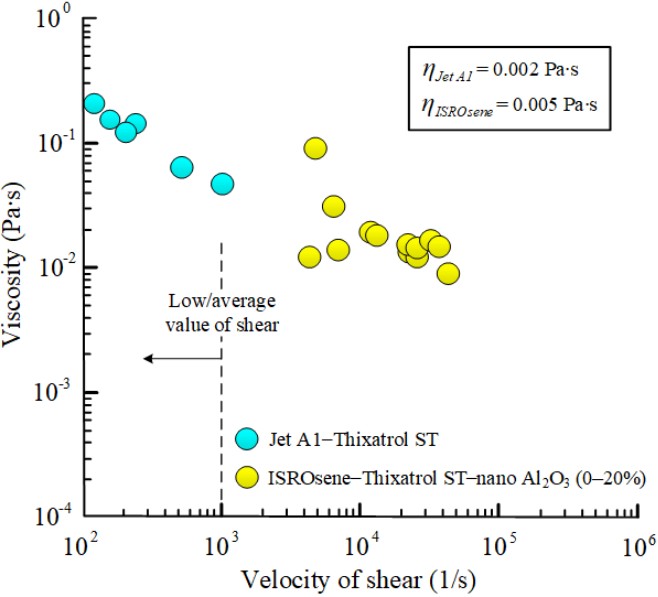

**Figure 3.** Viscosity change with shear rate under conditions of xylene gel (Jet A1 jet fuel, Thixatrol ST organic thickener) [113] and gel propellant with $Al_2O_3$ particles (ISROsene kerosene, Thixatrol ST organic thickener, xylene) [69,70] to spray device and during atomization.

The results [69,70] (Figure 3) illustrate that with increasing shear rate, the viscosity of xylene gel (Jet A1 jet fuel, Thixatrol ST organic thickener) decreases, and the viscosity of gel propellant (ISROsene kerosene, Thixatrol ST organic thickener, xylene) is quasi-constant. The spread of values according to the data (kerosene ISROsene, organic thickener Thixatrol ST, xylene) is due to different concentrations of 0–20% alumina particles in the fuel composition [69,70].

The shear rate arising from the movement of small fragments of fuel through the combustion chamber, as well as from the evaporation and combustion of fuel is very high (more than $10^5 \ s^{-1}$). The viscosity of a gel fuel under such conditions is several orders of magnitude greater than that of a typical Newtonian fluid, such as water [69,70].

### 3.2.2. Tensile Strength

The flow of gel fuels begins only when the limiting shear stress (tensile strength) is exceeded [113]. This means that gelled fuels liquefy under conditions where the shear stress exceeds the threshold value of tensile strength. It is known [126] that the tensile strength depends on the structure of the substance. Gel fuels, characterized by a high tensile strength compared to liquid fuels, are sprayed at the dispersion stage without sedimentation, which significantly increases the energy potential of the fuel. The results of direct measurements of the tensile strength, for example, using paddle mixers [127,128], depend on the sensitivity of the measuring device.

The tensile strength of gel fuels is not a constant value [129], as it depends on the registration time. There are studies [77,96,130], in which the tensile strength was determined using a conical penetrometer and a rheometer with an ascending sphere [130], as well as the minimum drop in capillary pressure at the moment of the start of liquefaction [77] and the maximum torque required to spin the rotor [96,130].

Using a Heavy Duty Shop Press 20 Ton hydraulic press, the tensile strength of gel fuels pellets (obtained after 15 cycles of freezing/thawing oil emulsions) with a characteristic size of 20 mm was measured for the following compositions group: 100–20 vol% PVA aqueous solution (5, 10 wt%) + 0–80 vol% oil. It has been established [109] that an increase in the oil concentration in fuel composition leads to a decrease in the tensile strength (Figure 4). At identical oil concentrations, the difference in the tensile strength of pellets based on 5% and 10% PVA solutions is 73–78% [109].

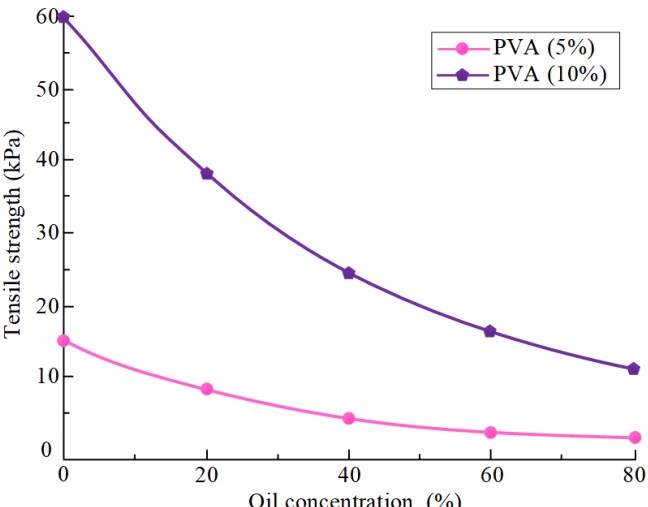

**Figure 4.** The dependence of the tensile strength of oil-filled cryogels fuel pellets on the oil concentration at different concentrations of PVA aqueous solutions [109].

The influence of elastic deformation along the axis of symmetry at the application of 6.5 kPa on the loss of moisture (oil) was established [109] for the same compositions. The authors found that the lower the density of the polymer matrix in the gel fuel, the greater the mass loss of the sample after deformation (Figure 5). The moisture loss of compositions based on a 10% PVA solution is on average 1% less than that of compositions with a 5% PVA solution. At higher oil concentrations, the density of the polymer matrix is lower. During deformation of oil-filled cryogels with an oil concentration of 80%, the loss of moisture due to deformation is 2–2.5 times greater than for cryogels containing 20% oil.

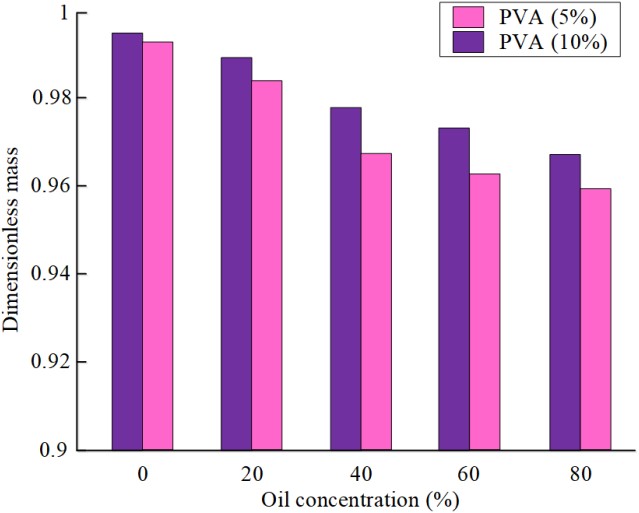

**Figure 5.** Masses of fuel pellets after deformation, reduced to their initial mass, at different concentrations of an aqueous solution of PVA [109].

In accordance with the technique described in detail in [110], the tensile strength of six compositions of oil-filled cryogels containing particles with different concentrations of coal dust with a dispersion of no more than 140 μm, as well as cryogels containing oil and coal particles, was studied. It has been established [110] that the adding of finely dispersed solid particles (up to 50%) into the cryogel composition leads to an increase in the tensile strength of fuel pellets by 1.7–2.3 times. Gel fuel pellets based on oil-filled cryogels with finely dispersed coal particles are characterized by lower elasticity and, as a result, higher tensile strength values compared to oil-free cryogel pellets.

### 3.2.3. Viscoelastic Properties

Gel fuels are viscoelastic substances that exhibit both viscosity and elasticity upon deformation [131]. Viscoelastic properties prevent leakage or spillage of gel fuels during storage and transportation, if the applied shear stress does not exceed the tensile strength. After deformation, the gel fuel partially or completely restores its geometric shape, if the tensile strength is not exceeded. Conventionally, gel fuels can be classified according to their viscoelastic properties into three groups: viscoelastic liquids, viscoelastic solids and elastic solids [131–134].

The Table 2 presents the results of studies of the viscoelastic properties of gel fuels and their analogues based on experimental studies of creep or oscillating shear.

**Table 2.** Viscoelastic properties of gel fuels and their analogues.

| Gel | Method | Notes | Ref. |
|---|---|---|---|
| 0.25–0.35% aqueous solution of Carbopol 934, stainless steel particles SUS304 (0–3 vol%) | resilience test after unloading; vibration shear test | similar viscoelastic properties of non-metallized and metallized gels; highly metallized gels are brittle | [91] |
| Liquid fuel Jet A1, organic thickener Thixatrol ST, xylene (85:7.5:7.5) | vibration shear test | dynamic modulus depends on frequency | [113] |
| Jet fuel JP-5, aluminum octoate (2–5%) | resilience test after unloading | viscoelastic solids; yield strength was not used | [77] |
| Nitromethane, 4–8% nanosized silica particles | vibration shear test | modulus of elasticity does not depend on frequency; the modulus of elasticity increases with increasing thickener content | [135] |
| 2% aqueous solution of polysaccharide | resilience test after unloading; vibration shear test | $\delta$ = 15–33° ($\omega$ = 0.1–10 Hz); verification of the Cox–Mertz relation was performed | [122] |
| Water, 4–6% hydroxypropylcellulose solution (monomethylhydrazine + hydroxypropylcellulose simulant) | vibration shear test | physical function does not depend on frequency; significant difference between $G'$ and $G''$ at $\omega$ = 0.2–10 Hz; the concentration of hydroxypropylcellulose does not affect $G'$ | [136] |
| 0.1–4% aqueous solution of Carbopol 941, triethanolamine | vibration shear test | compliance of microstructures with the frequency and range of voltage; triethanolamine contributes to the formation of fragile and rigid structures; the resulting gel (without triethanolamine) is Maxwell's fluid | [131] |
| Monomethylhydrazine, 6% aerosil; Monomethylhydrazine, 3% hydroxypropyl cellulose; Monomethylhydrazine, 2% hydroxypropyl cellulose + 3% aerosil | vibration shear test | hydroxypropylcellulose and hybrid gels are viscoelastic liquids and perfectly elastic substances; Aerosil based gels are viscoelastic solids | [133] |
| Water, 1% polysaccharide; water, 3% hydroxypropyl cellulose; water, Cekol 2000; water, Cekol 3000 | vibration shear test | type and concentration of the thickener do not affect $G'$; physical function dependent on frequency for hydroxypropyl cellulose and Cekol | [137] |

When conducting experimental studies on the creep of gel fuel, the shear stress is controlled (set) and the shear rate is changed in steps or in a continuous mode [138]. As a result, information is obtained not only about the viscoelastic properties of the gel fuel, but also about the creep yield strength of the gel fuel at a long-term constant shear stress [50]. The type of typical dependences obtained in the study of the creep of gel fuel [138] is shown in Figure 6.

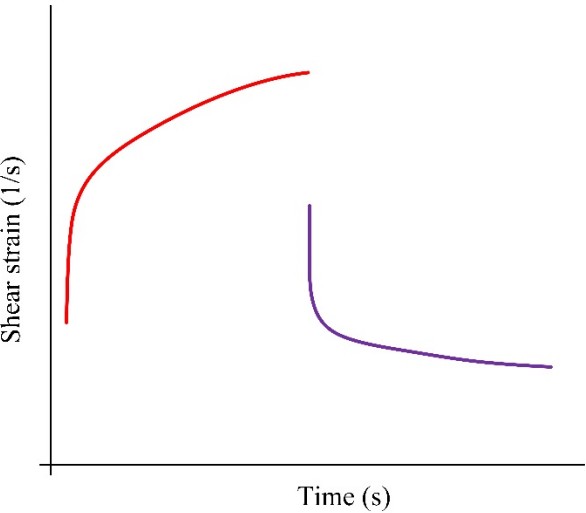

**Figure 6.** Type of typical dependences in the study of creep of gel fuel [138].

Experimental studies on oscillatory shear are carried out using rheometers or rheogoniometers. During dynamic (cyclic) tests, the gel fuel is subjected to a deformation that varies according to a sinusoidal law. Sinusoidal shear oscillations with a small amplitude are implemented using cone (or cylinder) oscillations, and the value of the deformation amplitude is fixed. The magnitude of stresses arising in the fuel is measured. Gel fuel is characterized by linear viscoelasticity if the resulting stresses change according to a sinusoidal law, but are out of phase with the deformation. The type of typical dependencies in the study of oscillatory shear is shown in Figure 7 [126,138].

The authors of [109] proposed a technique for studying the characteristics of elastic deformation of oil-filled cryogels based on 5% and 10% PVA aqueous solutions. It has been established that the higher the concentration of PVA in an aqueous solution and the lower the oil content in the fuel, the higher the elasticity of the pellets under otherwise identical conditions (Figure 8). The difference between the elastic modulus of pellets based on 5% and 10% PVA solutions with different oil concentrations is 27–71%. The results obtained qualitatively agree with the results in [139].

In accordance with the technique described in detail by the authors [109], the elastic deformation of heterogeneous gel fuels based on oil-free and oil-filled cryogels containing particles of coal dust with a dispersion of no more than 140 μm was studied [110]. It was concluded [110] that in the practical application of fuel pellets based on oil-filled cryogels, the necessary physical and mechanical characteristics (elasticity, tensile strength) can be achieved by varying both the concentration of PVA in an aqueous solution and oil in the primary emulsion.

It should be noted that currently, there are no known results of changing the viscoelastic properties of gel fuels or their analogues under conditions of ultra-high values of the cyclic frequency of shear deformation achieved during atomization in dispersing devices.

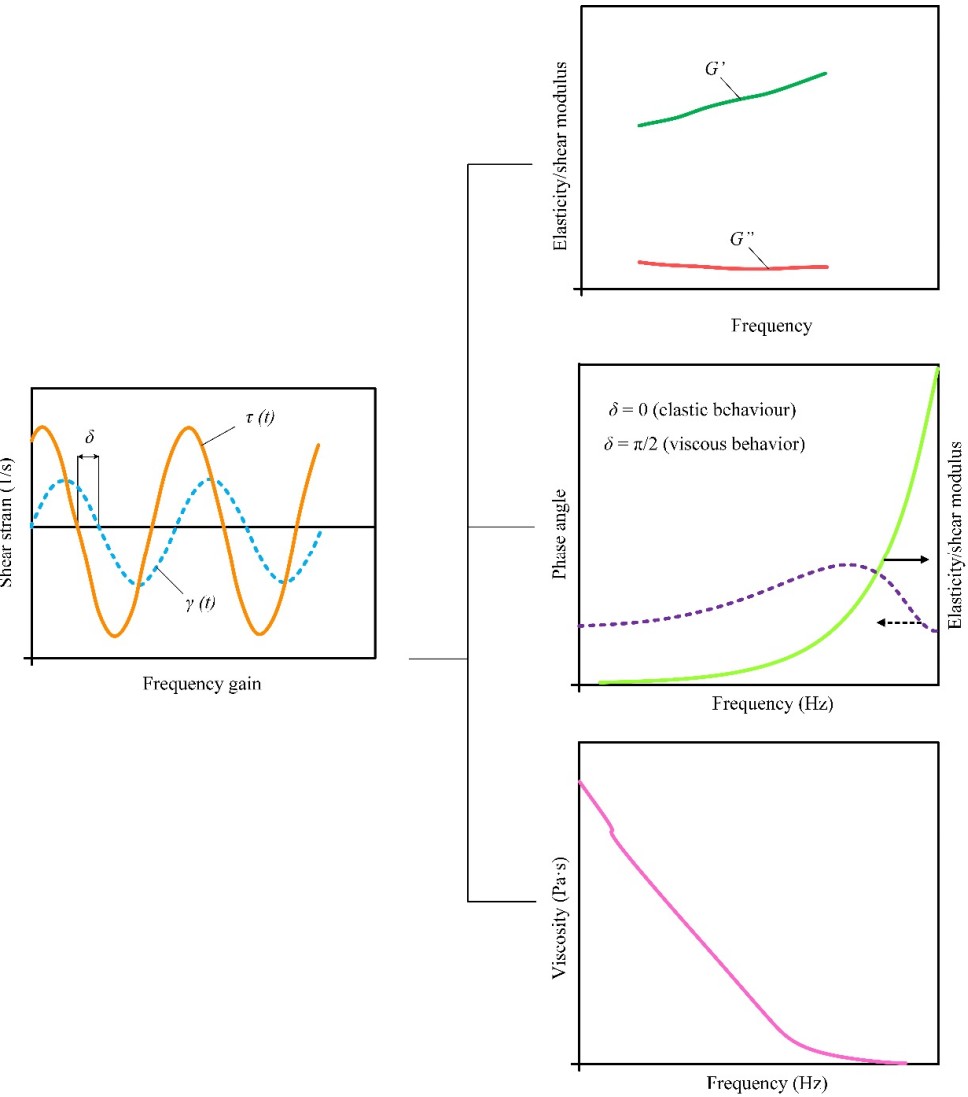

**Figure 7.** Typical trends obtained in the study of gel fuel for oscillatory shear [126,138].

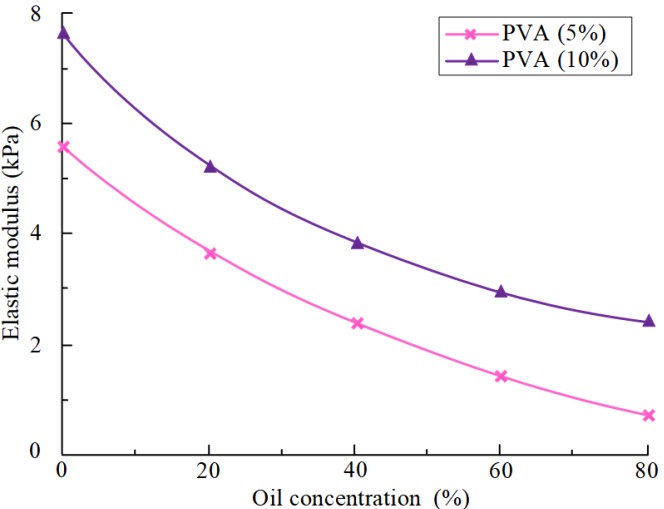

**Figure 8.** Elastic modulus of fuel pellets of oil-filled cryogels vs. oil concentration at various concentrations of aqueous solution of PVA.

### 3.2.4. Thixotropy

Thixotropy is a property of gelled fuels that is expressed by a decrease in consistency (thinning) when moving and an increase in consistency (solidification) when moving is stopped. Thixotropy depends on time and manifests itself due to structural changes at the microlevel. Under conditions of long-term storage of gel fuel [70], the consequences of thixotropy are a decrease in viscosity under constant shear conditions, as well as destruction of the structure (liquefaction). The most widely known are three methods for determining the thixotropy of gel fuels and their analogs: (1) "loop method" [35,77,132,140]; (2) by the change in dynamic viscosity over time [77,140]; and (3) "method of shear rate variation" [133].

Gel fuels are known, for example, to have a 3% aqueous solution of xanthan gum [140], for which the shear viscosity does not depend on time at a given shear rate. Such fuels liquefy without possible subsequent recovery (solidification). It should be noted that this property depends on the composition and its slight change leads to thixotropy, for example, as the 2% aqueous solution of xanthan gum is a thixotropic substance [141]. Other compositions that are characterized by a similar phenomenon are also known for example, a 0.3% aqueous solution of carbopol gel [141] and a 0.35% aqueous solution of carbopol gel [91]. It should be noted that there are gel fuels that are characterized by a complete restoration of the structure after shear termination, for example, D-gluconic acetals [142].

Thixotropy manifests itself insignificantly at high shear rates (more than $10^5$ s$^{-1}$) [132]. The influence of the composition of gel fuel on the thixotropy was studied [35,77,133]. It was shown [35] that the gel (unsymmetrical dimethylhydrazine, methylcellulose) becomes more thixotropic with an increase in the concentration of fine metal particles in its composition. It was also found [35] that thixotropy decreases with an increase in the temperature of gel fuels. A similar effect of temperature on thixotropy was reported by the authors [77].

When determining the thixotropy properties of gel fuels, the rate of recovery of the fuel structure after removal or reduction of shear is a more important characteristic than the rate of liquefaction. Restoration of the fuel structure is a longer process than dilution [126]. However, the structure recovery mechanism is currently unstudied.

## 4. Atomization

Fine atomization of gel fuel is necessary to achieve a high efficiency of the combustion process. The atomization of non-Newtonian fluids differs significantly from the atomization of Newtonian ones, but so far, there have been few results on the study of the influence of rheological properties on the atomization characteristics of non-Newtonian fluids [42,108,143].

When spraying gel fuels and films, a set of jets and droplets are formed, which contributes to an increase in the surface area. The latter makes it possible to intensify the processes of evaporation, mixing and combustion of the gel fuel. At present, a large number of different methods are known that make it possible to implement the spraying of Newtonian liquids. Most of them apply to gel fuels as well. The most widely used methods are based on the shock-jet mechanism of spraying (Figure 9). The latter is based on the collision of the two [68,69,71,88,89,99,100,135–137,141,143–163] or three jets [68,92,114,160,164,165].

When atomizing gel fuels by methods developed for atomizing Newtonian fluids, there are problems of delayed destruction of jets and the formation of relatively large droplets due to high viscosity [166]. One of the solutions to these problems is to improve the design of injectors and to overcome the viscous resistance of gel fuels by changing the flow pattern. A modernized design of the flash-boiling atomizer is known [166] (Figure 10).

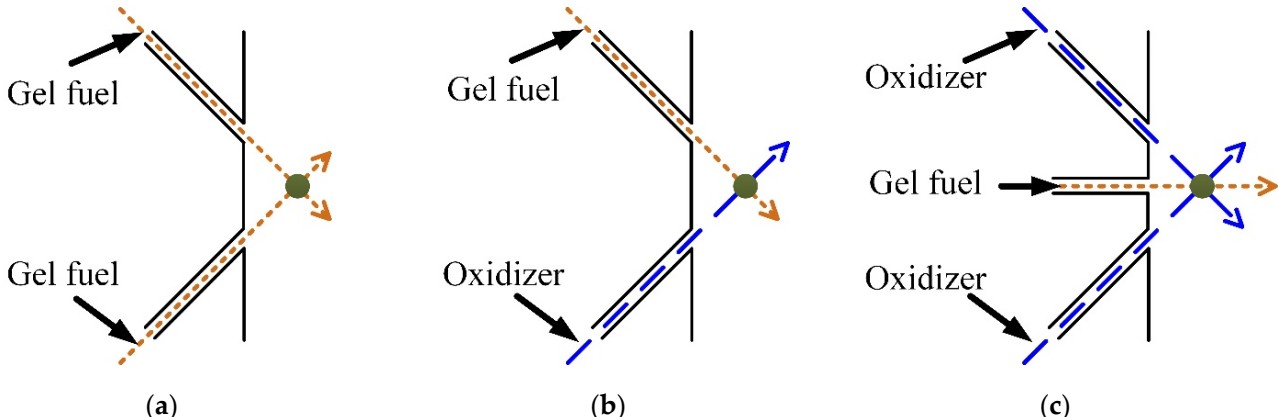

**Figure 9.** Schematic representation of two-jet (**a**,**b**) and three-jet (**c**) gel fuel atomization [92,99].

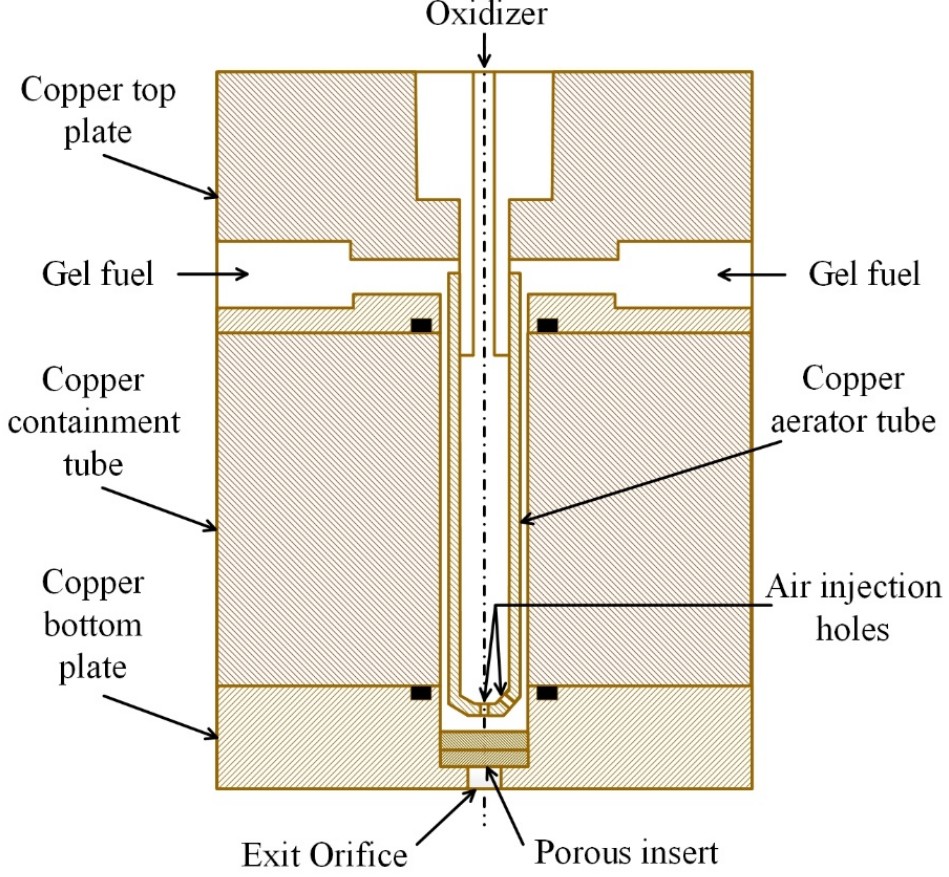

**Figure 10.** Schematic representation of the flash-boiling atomizer [166].

In Ref. [97], an improved three jet spray pattern (Figure 11) using two counter-rotating wheels was developed. Although this method is difficult to implement in practice, it shows that flow disturbance can be effective in overcoming the viscous (rheological) drag of gel fuels by modifying the atomizer design. A Helmholtz chamber with a jet drive was used to disturb the jet in the two-jet spraying scheme (Figure 11).

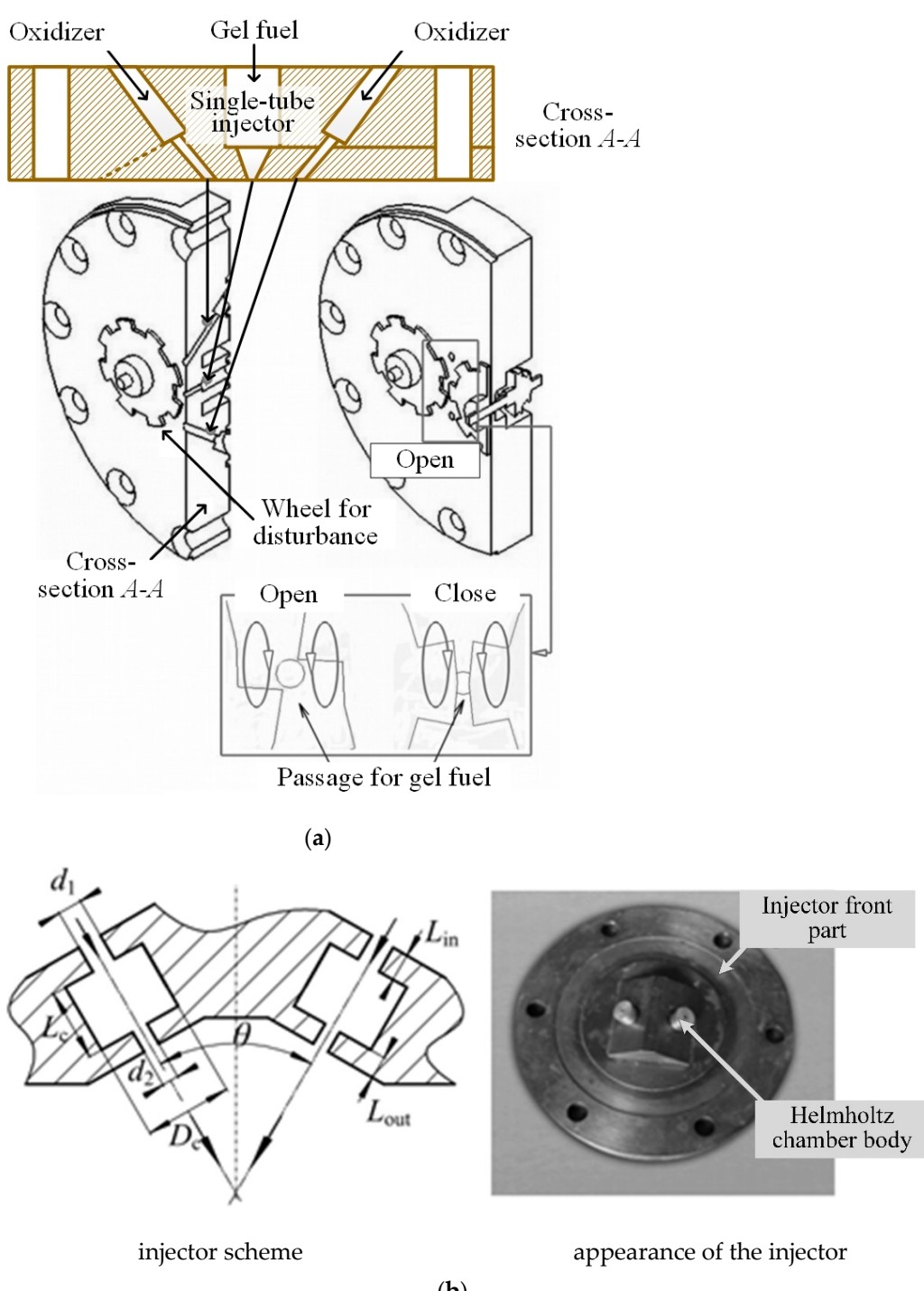

**Figure 11.** Schematic representation of an improved three-jet spray atomization (**a**) and a two-jet spray atomization with a Helmholtz chamber (**b**) [97].

### 4.1. Main Characteristics of Atomization

The analysis of video frames of high-speed video recording makes it possible to conduct a qualitative description of the mechanisms of destruction of gel fuel flows after their atomization from spray devices. Predictive assessment of the process of destruction of gel fuels is possible based on the results of an analysis of spray characteristics, which includes the breakup length ($L_b$), droplet size relative to known estimates of the average size ($D_{32}$) and spray cone angle ($\alpha$). Under conditions of shock wave formation, the frequency ($f$), length ($\lambda$), and wave propagation velocity ($u_p$) are evaluated.

#### 4.1.1. Breakup Length

The breakup length of the gel fuel spray membrane is the most important characteristic in determining the geometric dimensions of the combustion chambers. The results of studies of the breakup length of gel fuel and their simulants when flowing through a hydraulic nozzle [69,70,122,143,146,149,150,167,168] during vortex spraying [43,169] and two-jet spraying [69,70,143,146,149,150,167,168,170] have been established. Figure 12 shows the known results of flow breakup under the conditions of shock wave formation. It can be seen from Figure 12 that gels containing Thixatrol ST, as well as micro- and nanoparticles of aluminum and its oxide (Jet A1–7.5% Thixatrol ST–7.5% MIAK–35.4% m-Al and ISROsene–7.5% Thixatrol ST–7.5% xylene–(0–20% n-Al$_2$O$_3$)), are characterized by a significantly shorter breakup length than gels containing polymeric thickeners, carbopol, ethanol and xanthan gum. When spraying aqueous solutions containing more than 0.5% of carbopol thickeners and xanthan gum, the breakup length is longer compared to the breakup length of water. The addition of micro- and nanoparticles of metals to the composition of gel fuels reduces the breakup length, and an increase in the concentration of the thickener in the composition increases it.

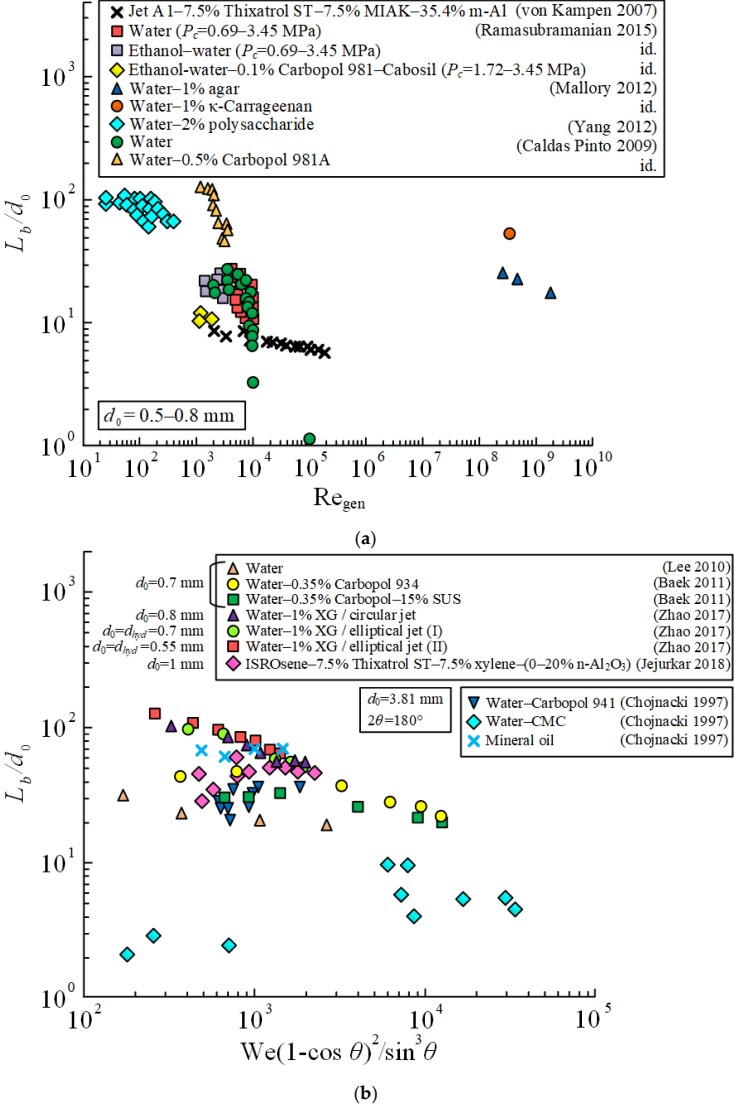

**Figure 12.** The breakup length of gel fuels and their simulants when flowing through a two-jet atomizer, depending on the: (**a**) Reynolds number [65,143,144,150,168,171]; (**b**) Weber number [68,69,149,167,170].

Under the conditions of vortex spraying of gel fuels, four typical regions were identified [172] along the flow breakup length (Figure 13): I—intact hollow conical sheet; II—the formation of holes connected by ligaments; III—turbulent flow of ligaments; and IV—breakup of ligaments into droplets.

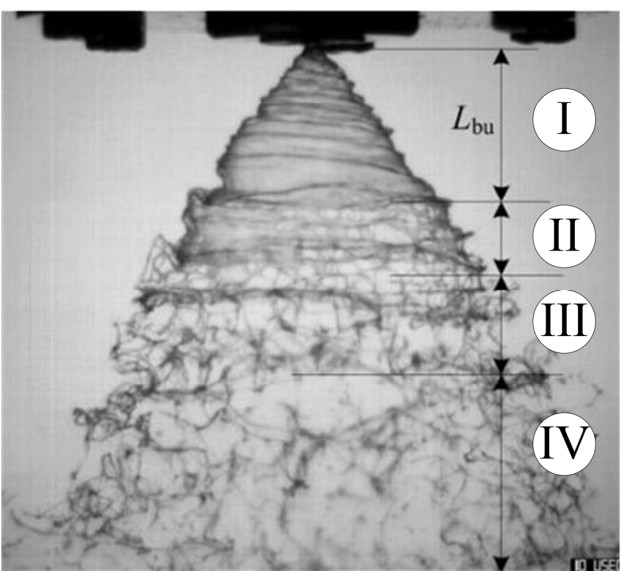

**Figure 13.** Typical regions of flow destruction along the breakup length [172].

In region I, the flow under the centrifugal force is transformed into a conical hollow thin layer. In region II, holes of a thin layer are formed, connected by ligaments. In area III, the thin layer is completely destroyed into ligaments. The latter, colliding with each other, form the structure of the "fluid web" [172]. In area IV, the ligaments form a structure in the form of "beads on a string" or partially collapse into droplets of irregular geometric shape. The length of the breakup of the gel fuel is determined by the length of the region I (Figure 13).

### 4.1.2. Spray Angle

Figure 14 shows the results of determining the dependences of the spray angle ($\alpha$) on the momentum flux ratio ($M$) [95,99], injection pressure ($P_{inj}$) [148], volume flow ($Q$) [173] and Weber number ($We_{gel}$) [68].

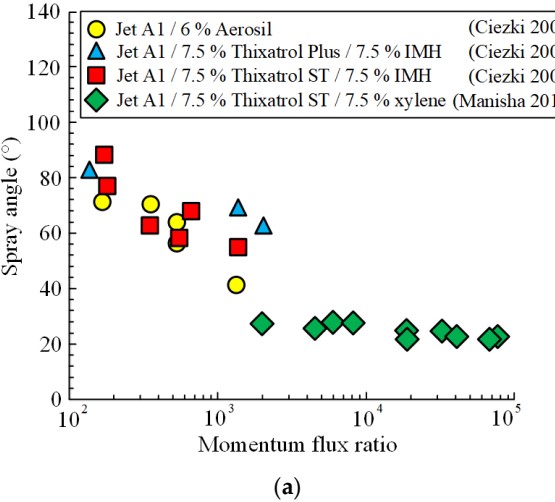

(a)

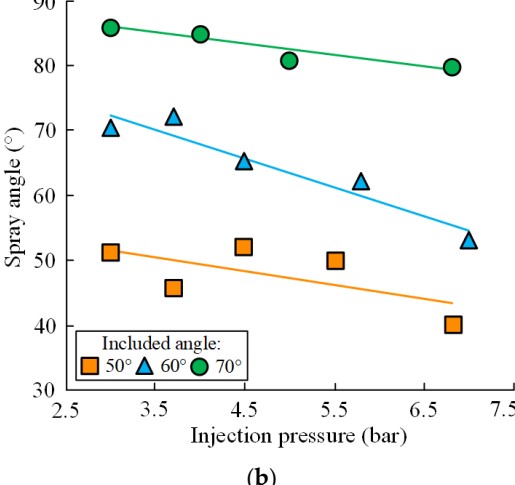

(b)

**Figure 14.** *Cont.*

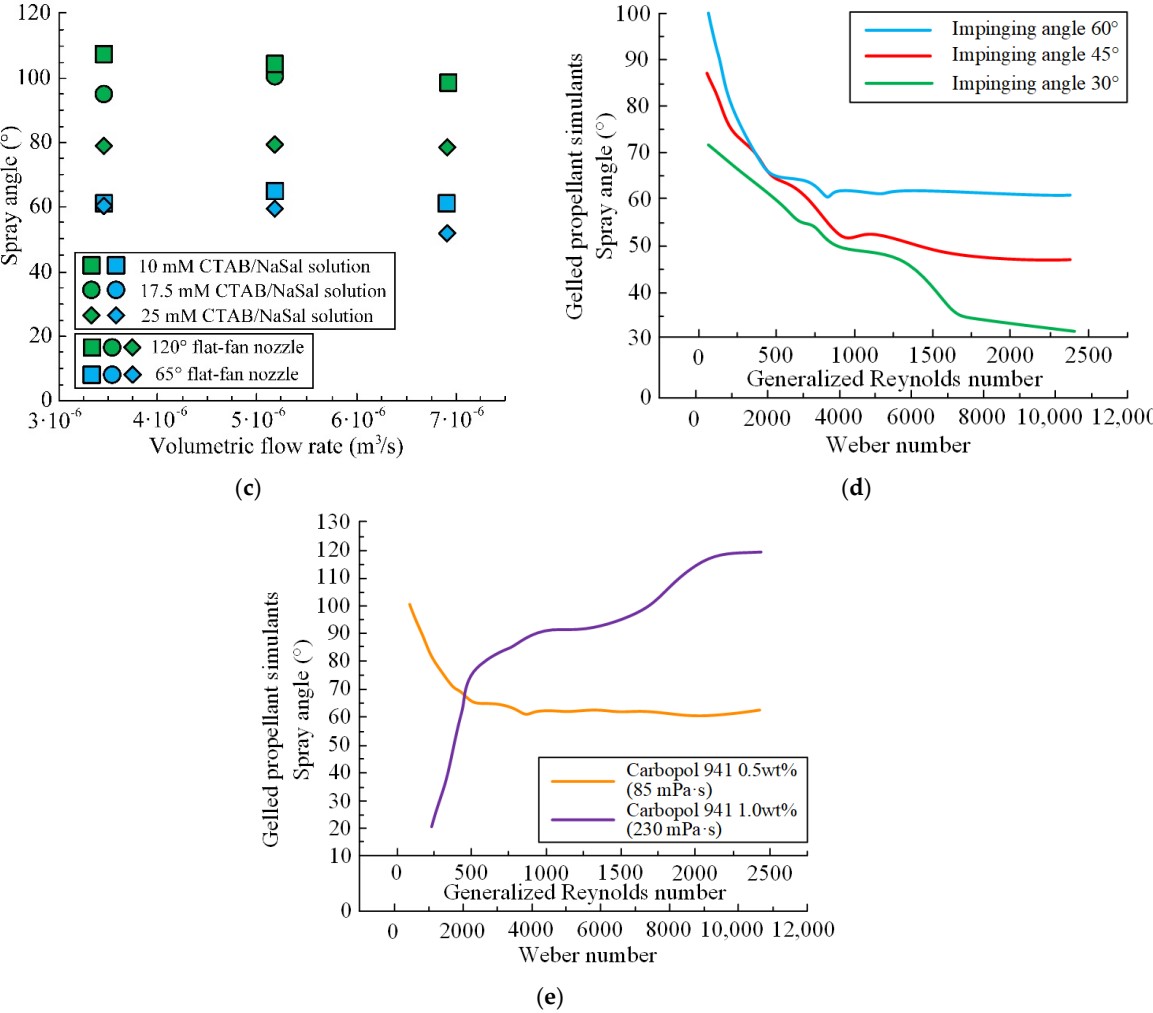

**Figure 14.** Spray angle versus: (**a**) Momentum flux ratio [95,99]; (**b**) Injection pressure [148]; (**c**) Volumetric flow rate [173]; (**d**,**e**) Weber and Reynolds numbers [68].

It is known [95] that the spray angle of the gel fuel decreases with an increase in the ratio of the mass flow rate of the gel fuel to a similar characteristic of the air flow, as well as with an increase in the viscosity of the fuel itself. In this case, the spray angle increases under conditions of a low ratio of the flow of gel fuel to air (2.7–4.0%). As this ratio increases (up to 8%), the spray angle decreases. The latter is due to the fact that at a ratio of the flow of gel fuel to air in the range from 4% to 8%, the viscosity increases, and the volume of the flow decreases [95]. At a low mass flow rate of the gel fuel, the cross-sectional area of the air flow increases, reducing the cross-sectional area of the fuel flow and increasing its speed. As a result, the layer decomposes into ligaments and further into droplets mainly in the axial direction. Consequently, the angle of the spray cone decreases. An increase in the ratio of mass flow rates of gel fuel to air flow by more than 8% does not affect the spray angle and remains practically constant within the high range of this ratio. The mass flow rate of the gel fuel is the dominant factor influencing the reduction in viscosity and thus reduces the spray angle as it increases. It can be seen (Figure 14a) that the composition of the gel fuel thickener has little effect on the spray angle [95,99].

Figure 14b shows the results of study [148] on the influence of pressure (from 3 to 7 bar, which corresponds to an average jet velocity of 10–20 m/s) on the spray angle. The experiments were carried out [148] with two-jet spraying of a gel fuel consisting of liquid kerosene (60%), aluminum particles 15 μm (30%), organophilic clay (8%) and propylene glycol (2%) at collision angles of 50°, 60° and 70°. It can be seen (Figure 14b) that the spray angle decreases as the injection pressure increases. As the injection pressure increases,

the impulse of the shock wave force increases and, consequently, the breakup length decreases [148].

The effect of volumetric flow rate on the spray angle (Figure 14c) was studied [106] by spraying solutions containing 10 to 25 mmol of cetyltrimethylammonium bromide (CTAB) and sodium salicylate (NaSal) in distilled deionized water through a stainless steel swirl nozzle with angle openings of 65° and 120°. It was found [106] that an increase in volume flow had little effect on the spray angle (Figure 14c). However, it was found [106] that an increase in elasticity greatly reduces the spray angle.

Figure 14d,e show the results [68] of determining the dependences of the spray angle on the generalized Reynolds number and Weber number under two-jet spray conditions. It was established [68] that the spray angle of gel fuels depends on the injection rate, jet collision angle and viscosity. With an increase in the injection rate of gel fuel, the spray angle decreases due to a decrease in the viscosity of the fuel due to stretching. As the Reynolds number increases, the spray angle approaches the jet collision angle (Figure 14d). Figure 14e shows that the curves characterizing the dependences of the spray angles of 0.5% and 1.0% aqueous solution of Carbopol with a viscosity of 85 mPa·s and 230 mPa·s, respectively, have different shapes. Under conditions of high-viscosity gel fuel atomization, the spray angle increases with an increase in the Weber number. Under conditions of low viscosity, the spray angle decreases and, at a Reynolds number of 570, tends to the jet collision angle [68].

### 4.1.3. Droplet Size Distribution

At present, a fairly extensive experimental database of statistical data on droplet sizes during the spraying of gel fuels and their simulants has been accumulated [69,144,148,149,174–176]. Figure 15 shows dependences of the ratio of Sauter mean droplet diameter to the injector outlet diameter ($D_{32}/d_0$) on the parameter ($d_0/u_j$) for gel fuels and their simulants [69,146,148,149,151,153,168,177] as well as water [146,148,149,168]. The parameter ($d_0/u_j$) characterizes the operation of the spray device, where $u_j$ is the velocity at the injector outlet. Based on the results of the analysis of Figure 15, it can be concluded that the diameter of droplets of gel fuels decreases to the size of droplets of a Newtonian liquid with a decrease in the diameter of the outlet of the atomizer and an increase in speed.

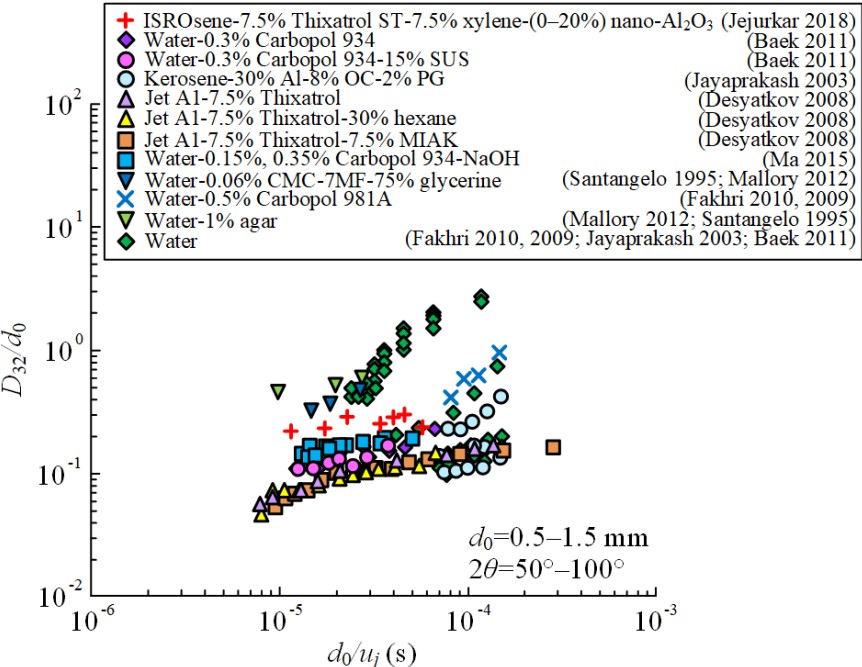

**Figure 15.** Dimensionless Sauter mean droplet diameters from the parameter of the spraying device ($d_0/u_j$) [69,146,148,149,151,153,168,171,178].

The droplet size can be reduced to 50 μm by changing the design of spraying devices [97]. Figure 16 shows the results of an experimental determination of the dimensionless Sauter mean droplet diameters, obtained with two-jet atomization (Figure 16) followed by a jet collision with the teeth of a pair of wheels (Figure 16a), using structures that initiate flow pulsations of three-jet spraying (Figure 16a) when using a porous insert (Figure 10) at the outlet of the "flash-boiling atomization" nozzle (Figure 16b) and atomizers with an internal impact on the flow (Figure 16c).

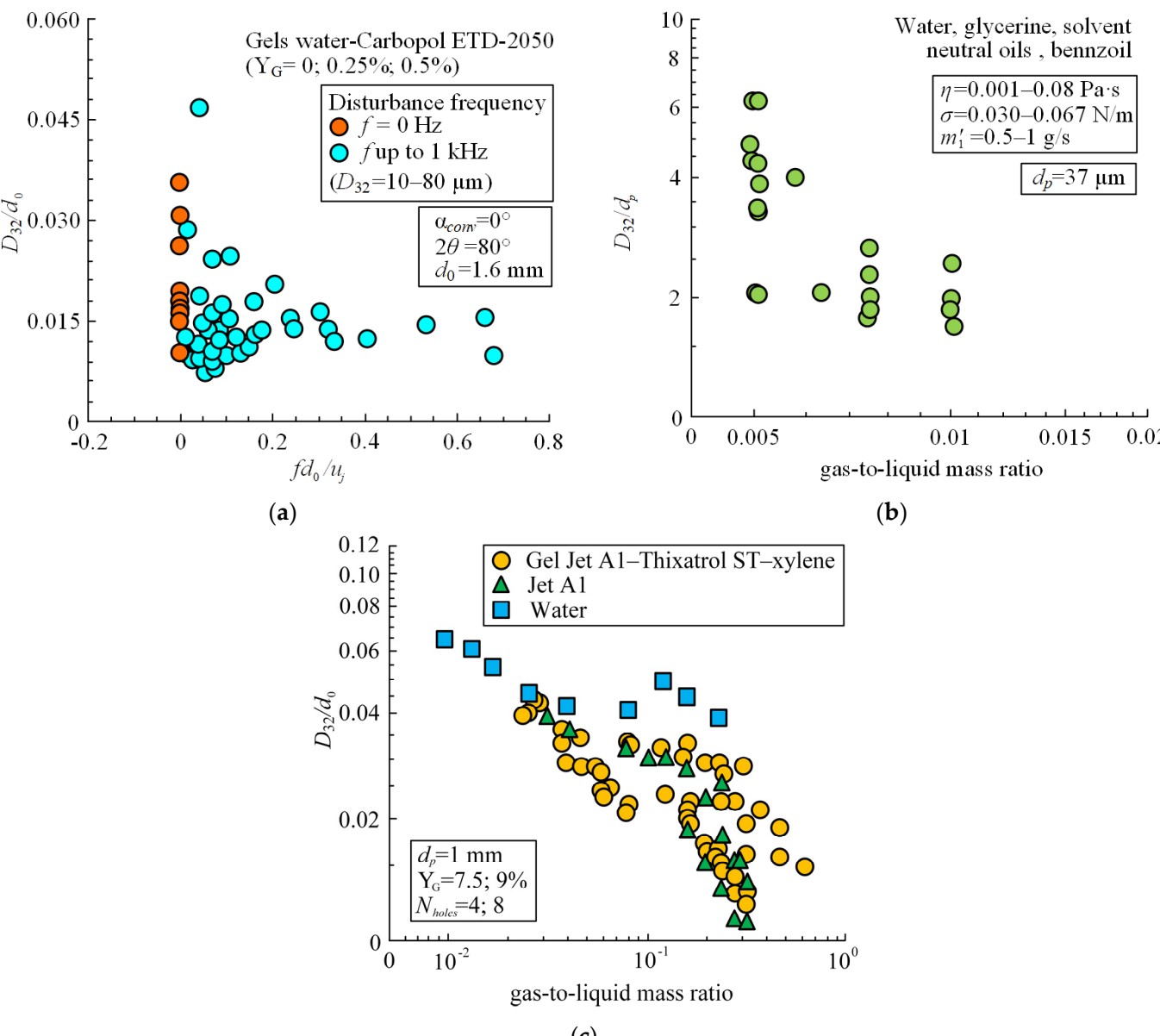

**Figure 16.** Dimensionless Sauter mean diameters of droplets. Designs of spraying devices: (**a**) Two-jet spraying followed by collision of the jets with the teeth of a pair of wheels and three-jet spraying with pulsating flows [97]; (**b**) "Flash-boiling atomizer" with a porous insert (pore diameter $d_p$) at the nozzle outlet [166,176]; (**c**) Atomizer with internal impact on flow [81,95,179].

It can be seen from Figure 16a that as the pulsation frequency of the three-jet spray and the wheel speed increase, the average droplet size decreases to values from 10 to 80 μm. The porous insert at the outlet of the "flash-boiling atomization" nozzle also significantly reduces the size of the resulting droplets (Figure 16b) while maintaining a low mass air

flow. When using atomizer designs with an internal impact, the average size of the formed droplets is in the range from 10 to 70 μm (Figure 16c).

Atomization efficiency mainly depends on the way the atomizing gas collides with the atomized liquid and the angle of their collision. If the jet collision angle is large, then the average droplet size does not depend on the diameter (thickness) of the gel fuel flow and the ambient pressure. If the collision angle is small (the air and fuel flows practically move in parallel), then after their collision, the average size of the droplets depends on the thickness of the fuel flow and the ambient air pressure. Atomizer designs that maintain prolonged contact between air and fuel, such as wave atomization and "flash-boiling atomization", perform more efficiently than designs based on two- and three-jet atomization.

## 5. Combustion

With the further development of the topic of gel fuels, such fuel can become an alternative to mixed solid fuel [180]. When using gel fuel, the same scheme of engine operation can be realized as when using high-power charges on an open surface. The aggregation state of gel fuel (depending on the surrounding conditions and on the purpose of the fuel) can change quite significantly – from liquid to solid elastically deformable The main difference between gel fuel and mixed solid fuel is the presence of the most complex physical and chemical processes occurring during its heating [180]. For example, when a mixed solid fuel containing ammonium perchlorate and butyl rubber is heated, the exothermic interaction of the fuel and oxidizer occurs in the near-surface layer of the heated region. Processes occurring during the induction period are well described in terms of the mathematical model of solid-phase ignition of condensed matter [181]. Gel fuels are characterized by gas-phase ignition. During fuel heating, a gas mixture is formed as a result of the evaporation of the fuel and mixing with an external oxidizer [48]. To predict the characteristics of gel fuel gas-phase ignition processes, appropriate mathematical models are needed, which differ, for example, from the models of ignition of solid [181] and liquid [182] fuels' condensed matter. The development of such predictive mathematical models is impossible without experimental investigations of ignition processes, which are a rather difficult task in the development of new fuels.

### 5.1. Experimental Investigation of Ignition and Combustion of Gel Fuel

Fuel particles were ignited under conditions of radiative-convective heating occurring in a high-temperature stationary air medium, which was generated in the volume of a ceramic tube (inner diameter 50 mm, length 450 mm) of a muffle furnace [183–185]. The range of air temperature variation in the furnace was $T_g$ = 700–1000 °C. After its stabilization, the fuel particle located on the holder was introduced into the cavity of the ceramic tube of the furnace using a minirobotic arm. The processes occurring during the induction period, $t_d$, were recorded by a color high-speed video camera.

Experimental studies [186] were carried out for two compositions of gel fuel based on an aqueous solution (10 wt%) of polyvinyl alcohol (PVA). The first composition (No. 1) was prepared without the addition of fine solid particles as follows: 50 vol% oil and 48 vol% aqueous solution of PVA and 2 vol% emulsifier. The second composition (No. 2) was prepared on the basis of the first by adding 30 wt% coal dust (140 μm) to the primary oil emulsion (50 vol% oil and 48 vol% PVA aqueous solution and 2 vol% emulsifier).

A fundamental difference was established between the mechanisms of ignition of single particles (melt droplets) of two compositions of gel fuels in a high-temperature stationary air medium [186]. Both fuels, when heated in the temperature range of 700–1000 °C, ignite with dispersion of melt droplets caused by the multicomponent composition of the fuel. During the melting of oil-filled cryogels, the liquid components of the fuel are separated. A shell of molten thickener is formed on the surface of the droplet. Under it is a flammable liquid. Because of such a structure of a droplet of molten gel fuel, when it is heated, processes occur that are not typical for the process of ignition of a droplet of a single-component combustible liquid, such as:

1. Formation of a large number of bubbles in the near-surface layer of the droplet;
2. An increase in the number and size of the bubbles, accompanied by a significant change in the shape of the drop;
3. Collapse of bubbles and dispersion of the initial droplet with the separation of a group of droplets of a significantly smaller size (1–2 orders smaller than the initial size of the fuel particle);
4. The previous process is accompanied by the release of combustible liquid vapor into the oxidizer medium through the thickener layer (outer shell) when the droplet is destroyed;
5. When the limiting conditions are reached, gas-phase ignition occurs in the vicinity of the droplet.

In contrast to liquid single-component fuels, the initiation of combustion of gel fuels occurs not in a small vicinity of a droplet, but in a rather large area (Figure 17), which has a positive effect on the development of subsequent combustion and intensification of the burnout of the components.

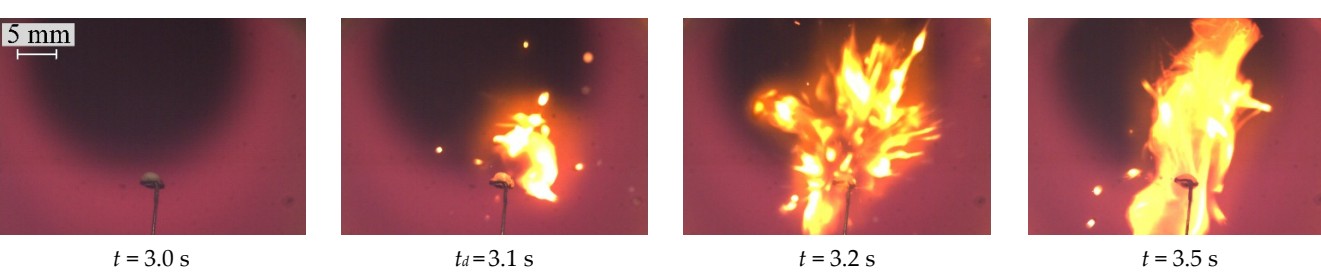

| $t = 3.0$ s | $t_d = 3.1$ s | $t = 3.2$ s | $t = 3.5$ s |

**Figure 17.** Typical videoframes of ignition and combustion of a particle of gel fuel (composition No. 1—without solid particles: 50 vol% oil + 48 vol% aqueous solution of PVA + 2 vol% emulsifier) at $T_g = 900\ °C$ [186].

The difference between the ignition mechanisms of these two gel fuel compositions is as follows. As a result of the collapse of bubbles and the dispersion of molten fuel droplet No. 2, vapors of a combustible liquid and fine coal particles, which are impregnated with this liquid, are blown into the high-temperature oxidizing medium. Therefore, gas-phase ignition of the gel fuel containing finely dispersed solid particles occurs more intensively and rather evenly in a large vapor-gas region (Figure 18).

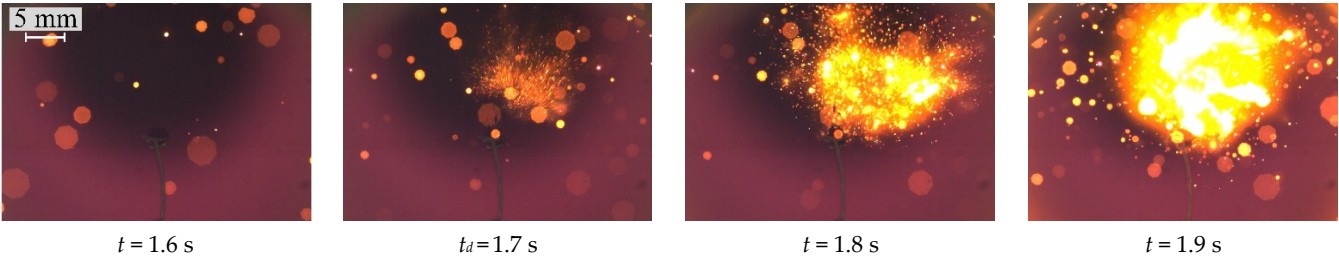

| $t = 1.6$ s | $t_d = 1.7$ s | $t = 1.8$ s | $t = 1.9$ s |

**Figure 18.** Typical videoframes of ignition and combustion of a particle of gel fuel (composition No. 2—with finely dispersed carbonaceous particles: 70 wt% (50 vol% oil + 48 vol% aqueous solution of PVA + 2 vol% emulsifier) + 30 wt% coal dust) at $T_g = 900\ °C$ [186].

By varying the temperature of the heated air in the range of 700–1000 °C and the velocities of entry into the combustion chamber of 0.06–0.10 m/s, the velocity (Figure 19a) and the size of the burnout area of fine fragments and the vapor-gas mixture after the dispersion of droplets of molten fuels were established (Figure 19b).

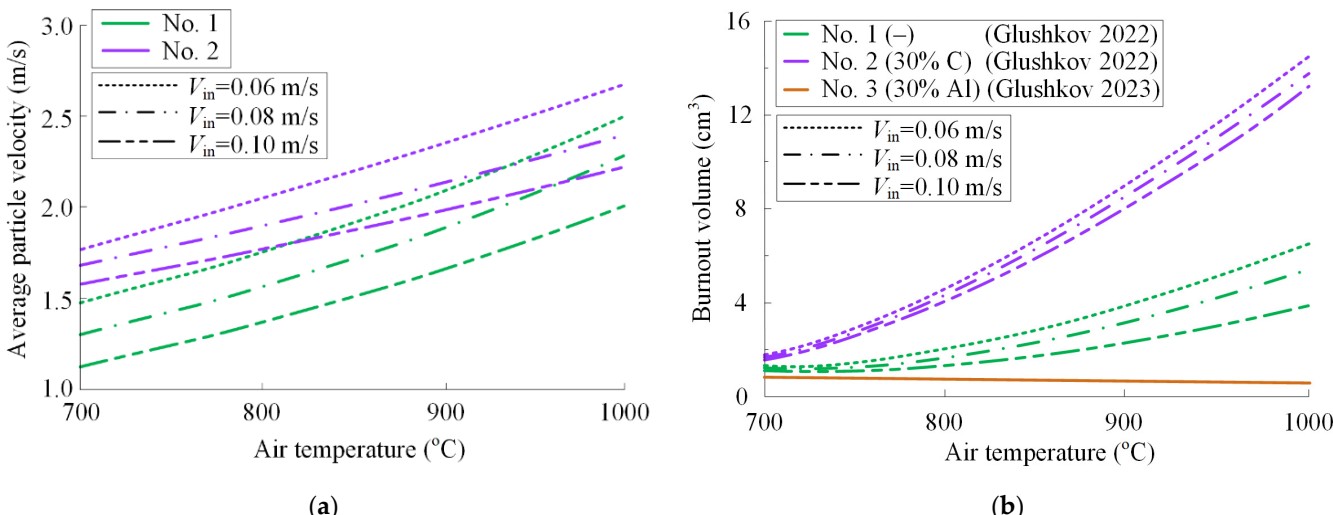

**Figure 19.** Characteristics of the dispersion of melts of gel fuels No. 1 and No. 2 at different rates of injection of fuel particles into the combustion chamber: (**a**) Speeds of movement of finely dispersed fragments during droplet dispersion [186]; (**b**) Dimensions of areas in which the process of burnout occurs [186,187].

Under such conditions, for an oil-filled cryogel without solid inclusions, the average velocities of fine particles after dispersion change from 1.1 to 2.2 m/s (by 90–100%), for a fuel composition, with addition of carbonaceous particles, ranging from 1.6 to 2.5 m/s (by 45–55%). The authors found (Figure 19b) two different trends in the dependence of the size of the burnout area on the temperature of the energy source. Compositions with a conspicuous gas-phase combustion without dispersion of a droplet melt (Figure 19b—No. 3: with the addition of 30 wt% V-ALEX 100 aluminum nanopowder to the primary oil emulsion) are characterized by a decrease in the size of the burnout region with increasing ambient temperature. Under radiant heating, after the droplet melts, the fuel evaporates monotonically according to the $d^2$ law. In the vicinity of the droplet, a spherical combustible gas-vapor mixture is formed, which ignites when the ignition temperature is reached. The higher the ambient temperature, the faster the necessary conditions for the ignition of the gas-vapor mixture are reached, the less time the combustible liquid has to evaporate and the smaller the size of the area in which the combustible gas-vapor mixture is formed. For fuel compositions that disperse during heating and combustion, the opposite trend is characteristic—the size of the burnout area increases with increasing ambient temperatures.

In this case, the energy release power of gel fuels is about 190 W in a volume of about 8 cm$^3$ with a burn-up of 10 mg of gel fuel (against about 25 W in a volume of about 1 cm$^3$ with a longer burning time of a single-component liquid fuel droplet).

Within the method of the two-color pyrometry [186], the temperature trends of the flame during the combustion of single particles of gel fuels were established. Fuel composition No. 1 is characterized by a series of episodes of dispersion after ignition of a fuel particle. The extremes on the temperature trends (Figure 20) correspond to the mechanism of ignition and combustion of the oil-filled cryogel described above, and illustrate a series of episodes of dispersion, proceeding until the complete burnout of the fuel components is achieved. The addition of fine combustible particles to the fuel composition not only increases the intensity of its ignition, but also ensures the occurrence of a fairly stable fuel burnout without significant temperature fluctuations during the combustion process (Figure 20). The result of experimental studies indicates that the regularities and characteristics of the ignition of gel fuels (both with the addition of solid particles and without them) can vary greatly. Such a wide variety of combustion mechanisms and characteristics is one of the advantages of using gel fuels compared to liquid and solid fuels.

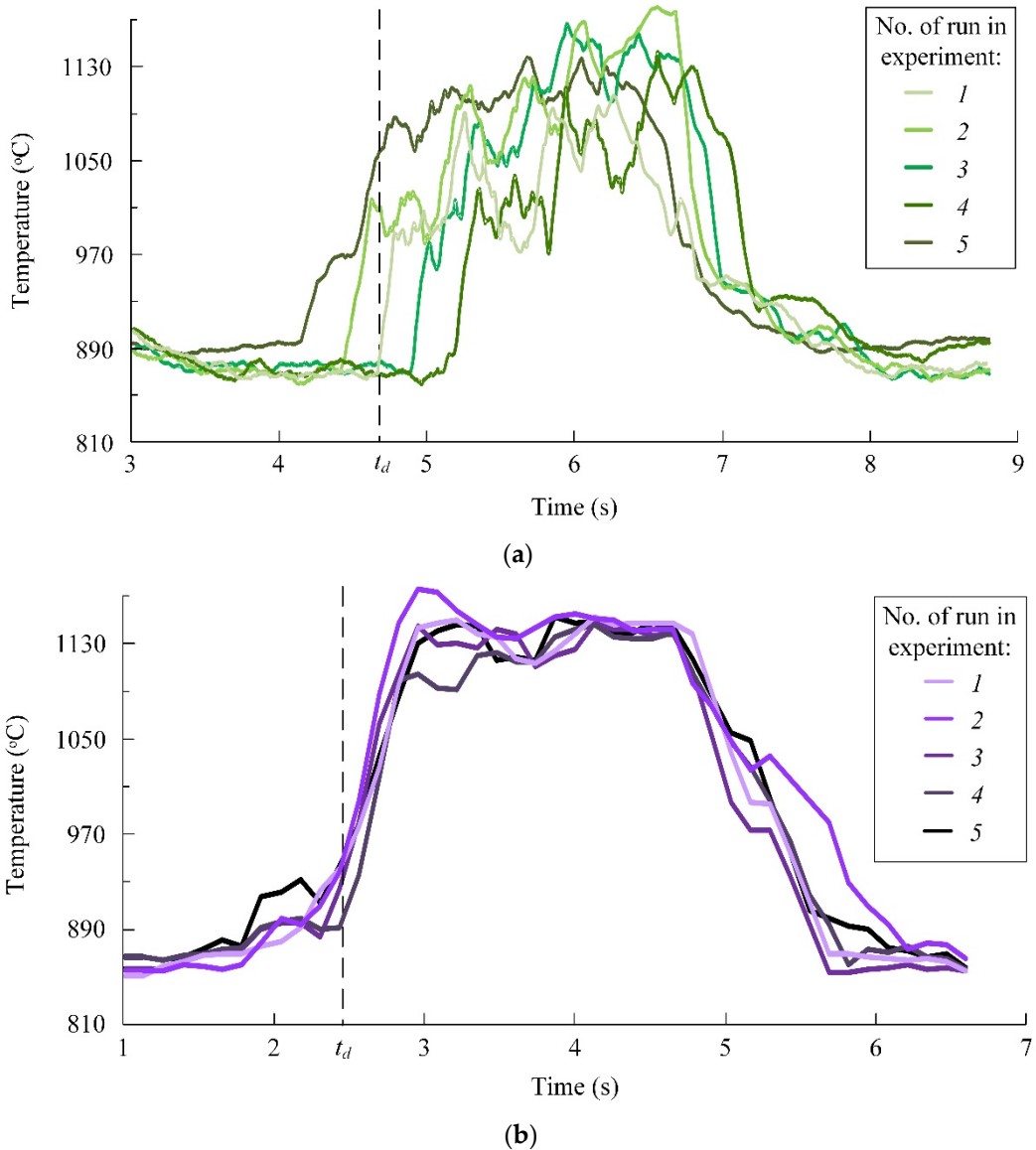

**Figure 20.** Flame temperature during the combustion of gel fuel at $T_g$ = 850 °C: (**a**) No. 1 and (**b**) No. 2; *1–5*—number of run in a series of experiments.

### 5.2. Mathematical Models of Ignition Processes

Based on the results of experimental studies [109,188,189], a scheme was developed (Figure 21) for the ignition of a typical gel fuel particle based on an organic polymer thickener in a high-temperature air medium. It is assumed [109,188,189] that a cold fuel particle ($T = T_0$) of a spherical shape with a radius $r_p$ at the initial time ($t = 0$) is introduced into the region of still air heated to high temperatures ($T = T_a$) (Figure 21a). When the particle is heated (mainly due to the radiant heat supply), its surface temperature reaches the melting temperature of the gel fuel ($T_{melt}$ = 330 K) and a melting front is formed, the boundary of which moves towards the center of the particle (Figure 21b). After the gel fuel is melted, its components are almost instantly separated. A thin thickener layer is formed on the surface of the melt, under which a combustible liquid is located (Figure 21c). The thickness of the thickener layer decreases until it completely evaporates (Figure 21d). After that, the combustible liquid begins to evaporate from the free surface (Figure 21d). In the vicinity of a fuel particle, a combustible gas-vapor mixture is formed as a result of diffusion (Figure 21e). When limiting conditions are reached, gas-phase ignition occurs (Figure 21e).

The event of excess temperature in the zone of exothermic reaction of the temperature of heated air ($T > T_a$) is accepted as the ignition criterion [109,188,189].

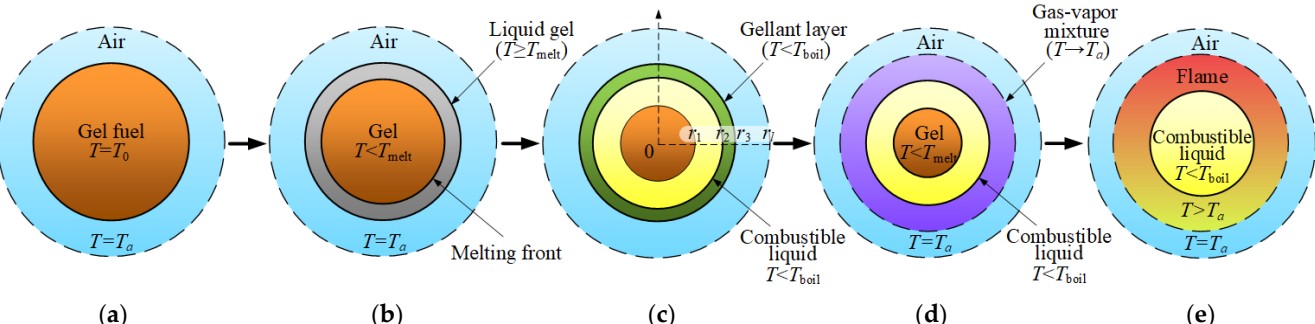

**Figure 21.** Scheme of the ignition process of a gel fuel particle [109,188,189]: (**a**) initial moment ($t = 0$); (**b**) gel fuel melting; (**c**) separation of gel components; (**d**) formation of combustible gas-vapor mixture; (**e**) ignition moment ($t = t_d$).

As a result of numerical simulation of the ignition process [109,188,189], temperature profiles in the fuel particle/air system (typical ones are shown in Figure 22) at different times during the induction period $t_d$ with the initial particle radius $r_p$ = 1.25 mm and heated air temperature $T_a$ = 1200 K. The inflection points on the temperature profiles at $0 < r \leq r_p$ correspond to the coordinate of the fuel melt droplet surface. The concentration of the thickener (water solution of PVA) in the composition of the gel fuel was $\mu_4$ = 48%.

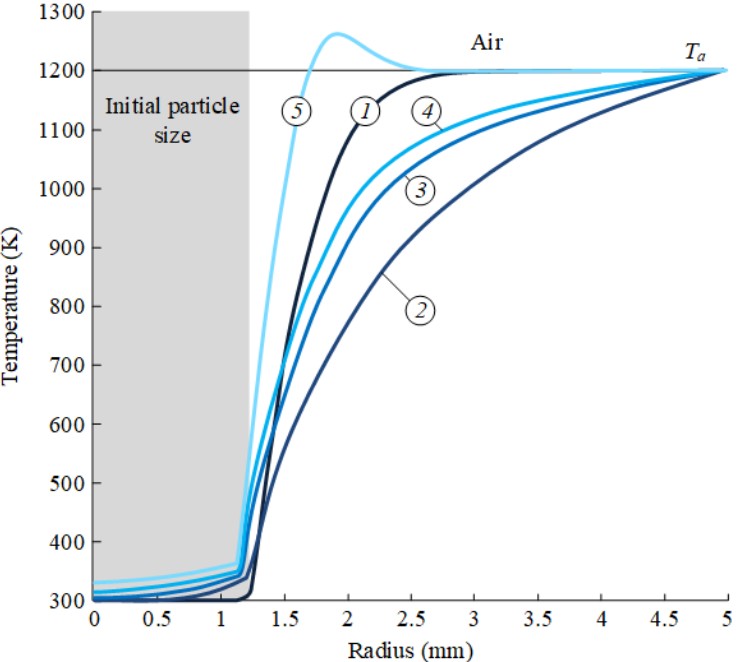

**Figure 22.** Temperature profiles in the fuel particle/air system at $T_a$ = 1200 K, $r_p$ = 1.25 mm, $\mu_4$ = 48% at different time points [189]: *1*—$t$ = 0.1 s; *2*—$t$ = 0.386 s; *3*—$t$ = 0.77 s; *4*—$t$ = 1.16 s; *5*—$t$ = 1.55 s.

The induction period $t_d$ can be conditionally divided into two stages. The first stage is the predominance of heat and mass transfer processes during the heating of a fuel particle, the formation of a combustible gas-vapor mixture and its heating. The second stage is the predominance of chemical reaction processes during the interaction of the components of the gas mixture. The characteristic times of these stages differ significantly. The duration of the first stage is 70–90% of the time $t_d$ (depending on the temperature of the heated air). The second stage is much more intense and its duration depends mainly on the kinetic

parameters of the process of chemical reaction of the combustible component and the oxidizer. Therefore, the processes of heat and mass transfer have a rather significant effect on the intensity of ignition of the fuel as a whole.

It has been established [189] that the temperature $T_a$ = 800 K is the minimum required for ignition of a gel fuel particle. At lower air temperatures, the fuel does not burn. Under such conditions, the energy of the heating source is not enough to ignite the resulting gas-vapor mixture. The results of numerical simulation are in good agreement with the experimental data [109] obtained under identical conditions. The difference in ignition delay times does not exceed the random error (20%) of recording $t_d$ in experiments. [109]. The theoretically established fuel ignition delay times in the heated air temperature range of 800–1473 K [189] are reduced by more than six times (Figure 23). When close to the limiting conditions of ignition (when $t_d$ = 8.15 s), the duration of the induction period, $t_d$, asymptotically tends to infinity at $T_a{\rightarrow}$800 K (Figure 23). Under conditions of high ambient temperatures (at $T_a$ > 1150 K), when physicochemical processes reach their maximum intensity, the fuel ignition delay time does not decrease to less than 1.3 s (Figure 23). Thus, for ignition conditions, the preparation time for the fuel particle/air system for the development of an intense exothermic reaction process is about 1.2 s at an oxidizer temperature exceeding 1150 K. As a rule, in practice, the fuel enters the combustion chamber in the form of a flow of particles which sizes may vary within a certain range. Therefore, for the possible practical application of the results of studies [189], a more detailed analysis of the influence of this factor on the ignition characteristics of gel fuel is required.

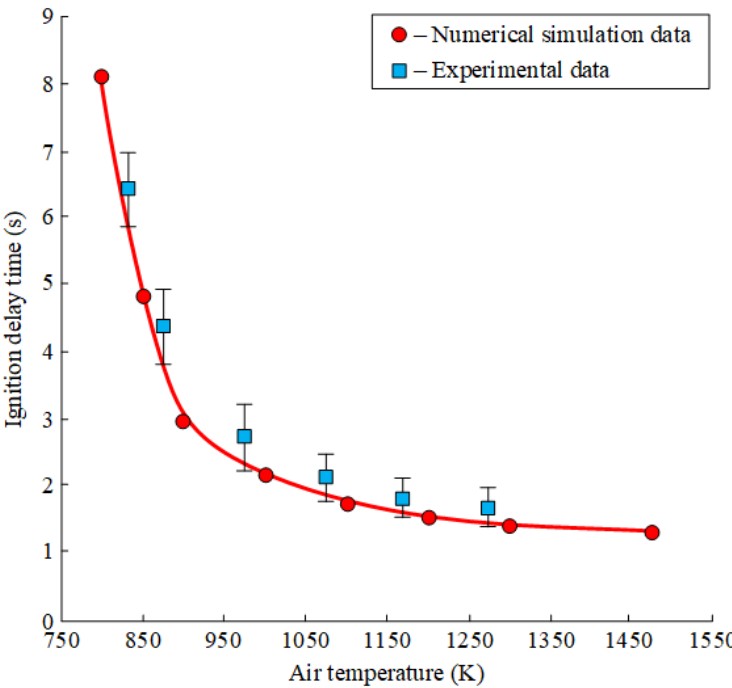

**Figure 23.** Dependence of the ignition delay time of gel fuel particles on the temperature of heated air at $r_p$ = 1.25 mm, $\mu_4$ = 48% [189].

## 6. Conclusions

Over the past two decades, a fairly large amount of new knowledge about gel fuels has been obtained within the previously identified areas of research on fuel component composition, rheological characteristics, atomization and combustion processes [24,50,55,64,65,70,81,91,94,100,115,144,149–151,190–192]. In addition to laboratory studies, full-scale tests of the first rocket engines operating on gel fuels were carried out. Future advances in this area contributing to the emergence of mass-produced engines will depend on success in the implementation of research and development work in the

following areas, consistent with the global trend in the development of fundamental and applied research on gel fuels.

### 6.1. Fuel Composition

The difficulties in optimizing high metal content (up to 60%) in gel fuels require the increased attention of the scientific community. Studies have shown that promising gel fuels with a high content of fine metal particles behave like unstable slurry fuels. Experience with the use of liquid fuels shows that the metal content can reach 60%. This result should be replicated for gel propellants as part of a large-scale research cycle.

Considering the direct connection between the component composition and rheology (as well as the atomization process), it is advisable to optimize fuel compositions iteratively, including rheology and atomization data in the corresponding search algorithm.

In the future, the plan is to apply systematic multi-criteria methods for searching for optimal gel fuel compositions.

Previously, large-scale studies of compositions in the selected areas in the framework of the study of gel fuels, including metallized ones, were not carried out. Thus, there is no information on fuels' stability and the influence of the processes of components settling and separation on their uniformity of composition, rheological characteristics, atomization and combustion. Most of the results were obtained for fuel samples weighing less than 1 kg, or in some cases for samples weighing 1–10 kg. The influence of the scale factor on the quality of the prepared gel fuel is unknown.

### 6.2. Rheology

The effect of tensile strength, thixotropy and viscoelasticity on failing structures remains to be seen. This area of research should be expanded by considering secondary decay events.

The rheological properties of "brittle" or "extremely fluid" gels can play a decisive role in their spray characteristics. Since gels with a high concentration of the metal component can be "brittle", it is important to study the correlation between the destruction processes of such gels and their rheological properties.

Research should be carried out on methods of viscosity reduction in laboratory and in-line measurements to increase viscosity reduction beyond what is currently achieved by shifting fuel delivery and injection systems. The results of such studies should be useful for the effective design of appropriate systems for aerospace industry objects with limited weight and size characteristics.

### 6.3. Atomization

The destruction modes of gel fuels during spraying should be systematically checked to identify the influence of rheological properties.

Original approaches to the development of the design of advanced injectors should consider the rheological behavior of gel propellants. This will not only effectively organize fuel atomization in the combustion chamber, but also maintain its combustion characteristics at a predetermined level.

The feature of gel fuels containing particles of thickeners (especially silica) and metal particles has been reliably established—to form fragments of irregular shape during spraying. Further dynamics of changes in the shape and size of these fragments and their effect on the flame should be studied in more detail. Approaches to the formation of a component composition that combine two types of thickeners, one of which is polymeric, should prove useful in balancing the tendency to form irregular fragments and relatively easy fuel atomization.

It is known that the secondary dispersion of metalized liquid fuel droplets is critical for the ignition and combustion of metal particles. It is assumed that similar problems will arise for metallized gel fuels, which needs to be confirmed.

### 6.4. Combustion

Promising mathematical models of the combustion of gel fuel single droplets based on a polymer thickener should describe the physical processes that lead to the oscillatory nature of the injection of combustible liquid vapor into the oxidizer medium and the formation of a vapor-gas mixture and its burnout. To develop the corresponding mathematical models, a theoretical basis is needed in the form of the results of experimental studies cycle using high-speed video recording and non-contact diagnostics of gas flows.

A detailed understanding of the regularities and characteristics of physicochemical processes occurring during flaring of gel fuels is necessary to overcome the gap between theoretical and empirical knowledge in order to increase the reliability of known predictive models describing the corresponding processes.

Experimental data on the characteristics of metallized gel fuel flare combustion have not yet been published. The results of fundamental research are important for carrying out development work in the design of a ramjet engine.

Research is required to determine effective solutions to reduce fuel viscosity in a short period of time, reduce the overall dimensions of the nozzle while maintaining its performance, increase the degree of fuel metallization and control the dynamics of solid metal oxide particles and the processes of their interaction with the combustion chamber walls and nozzle.

**Author Contributions:** D.G.—writing, review and editing; K.P.—writing, review and editing and A.P.—writing, review and editing. All authors have read and agreed to the published version of the manuscript.

**Funding:** The reported study was funded by the Russian Science Foundation grant number [18-13-00031], URL (accessed on 26.12.2022): https://rscf.ru/project/21-13-28043/.

**Conflicts of Interest:** The authors declare no conflict of interest.

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
