# Peer review of "Gel Fuels: Preparing, Rheology, Atomization, Combustion"

_energies, doi:10.3390/en16010298_

Round 1

Reviewer 1 Report

The review paper entitled "Gel fuels: preparing, rheology, atomization, combustion" describes the preparation and processing of gel fuels as a replacement for air-jet fuels. The topic is highly valid, however, I found this paper not written precisely as it is suggested for a review.

First of all, some citations given are not necessary. Perhaps, table 1 shows authors working on the subject of gel fuels. I do not find mentioning their names in scientific data. Rather than that, their achievements could be presented.

The descriptions of gel fuel properties are given very briefly. For example, the authors tell that metal particles are added to gels, but there is no information on what kind of metals are they. Why are they selected? At fig. 2. the combustion stages of gels are given, however, it is not described at all. There are many times given comments that some values are high or low, however, compared to what? How much/many are they?

There are several comments given that 'data is given in the reference' and a list of 10 literature sources is given. Here, I recommend discussing the obtained results and telling what is the general outcome. On the other hand, not everywhere the reference is needed, perhaps while describing what kind of fuels we recognize (solid, liquid, gases) or what is used to analyze viscosity.

Please avoid statements that 'various properties were studied by various scientists.'

Kind regards.

Author Response

Please find attached file with the detailed response to the Reviewer 1.

Reviewer 2 Report

The proposed manuscript deals with Gel fuels.

The advantages and the achievements of gel fuels are clearly underlined in the Introduction.

The search and the consultation of the published papers and documents relevant for the subject turns out numerous, various and adequate to a review job.

The review is comprehensive and considers the whole development history of gels and their adaptation to fuels, going back and starting from pioneeristic works, as early as 1936 [Ref. 77].

Conclusions and Recommendations are comprehensively and systematically addressed by category with ref. to the subject.

MAJOR ISSUES

Previous review on the subject

Padwal et al. published in 2021 a comprehensive review under the title “Gel propellants” in:

Progress in Energy and Combustion Science, Volume 83, 2021, 100885, ISSN 0360-1285, https://doi.org/10.1016/j.pecs.2020.100885

which is not quoted here.

Therefore, the Reviewer asks the Authors to include this published review within the papers addressed in “Introduction” and to explain where/how this newly proposed review is differentiated or improved with respect to Manisha et al. (2021)

Self-references

There many self-references, e.g., to the principal investigator: D. Glushkov.

The Reviewer counted as many as 15 self-references.

Although the principal investigator dr. D. Glushkov may be an expert in the field and has developed/published a lot of work, the Reviewer strongly believes that the number of self-citations is to be reduced. 

Therefore, the Reviewer asks the Authors to manage a possible and reasonable reduction in the number of self-citations. 

Technology

When the Authors indirectly refer to the technology for atomization, at either lab scale or proven application, the Reviewer – as a consequence the reader – expect to find at least a schematic representation of nozzle configurations.

The Reviewer asks that at least a new Figure is introduced and assists the description and the understanding of sentences that describe the results like these ones:

The most widely used methods are based on the shock-jet mechanism of spraying. The latter is based on the collision of two or three jets

three-jet spraying (Figure 10a), when using a porous insert at the outlet of the "flash-boiling atomization" nozzle (Figure 10b) and atomizers with an internal impact on the flow (Figure 10c).

Sauter mean droplet diameters, obtained with two-jet atomization followed by a jet collision with the teeth of a pair of wheels (Figure 10a),

Figure 14

The meaning and the outcome of the results are unclear. 

Fuel composition, No.1 and No.2, numbers from 1 to 5 in the legend are unexplained or unclear.

MINOR ISSUES
row

19-20     The Reviewer (and the future reader) expects to find here for “liquid [1,2], solid [3,4] and gaseous [5,6] fuels” very general and wide-scope references (e.g., books) and NOT very specific research articles like those cited under [1 to 6]; for instance, the Authors have done this for ref [27]: “Williams, F.A. Combustion Theory; 2nd ed.; Westview Press: Boulder, United States, 1985”.     -->  Please change Ref. under [1 to 6] !

33   coal-water slurries [16], coal-water slurries with petrochemicals [17]    -->  Also here, please change or add a ref. to a more general and wide-scope publication: “Miccio Francesco, Raganati Federica, Ammendola Paola, Okasha Farouk, Miccio Michele. Fluidized Bed Combustion and Gasification of Fossil and Renewable Slurry Fuels. Energies 2021, 14, pp. 7766-7781, https://doi.org/10.3390/en14227766“

60   “at a force of 1 kg per 1 kg of fuel”    -->  convert to SI units

445  Figure 6: the Reviewer doesn’t find “… gels containing silica thickener particles, as well as micro- and nanoparticles of aluminum and its oxide” in the legend

463-464   Figure 8c: volumetric flow rate is expected from both text and legend, whereas the x-axis caption is “Momentum flux ratio”      -->  pls. check !

463-464   Figure 8c: the values reported on the x-axis scale for the volumetric flow rate (from 3 to 6*10-6) appear too low to the Reviewer   -->  pls. check !

513  “uis the flow rate at…”      -->  “uj is the velocity at…”

577  “the induction period were…”  -->  “the induction period td were”

Figure 10b          what is dp in the y-axis title?

Figure 10b and c    what is GLR in the x-axis title?

Figure 11           Pls. either report these results as work carried out by the Authors or cite the investigators as a Ref. 

Figure 12           Pls. either report these results as work carried out by the Authors or cite the investigators as a Ref. 

643  “Speeds of movement finely dispersed…”   -->  “Speeds of movement of finely dispersed…”

Figure 13b     if the y-axis title is “burnout area”, then its units are “cm2” -->  pls. check !

692  “The induction period can. ..”   -->  “The induction period td can...”

ENGLISH TEXT 

Perfect spelling and excellent fluency.
Here are some minor corrections:
row

74   “…is a comprehensive study of gel fuels.” -->  “…is a comprehensive review of studies on gel fuels.”

326  “at different aqueous solution of PVA concentrations [125].”   -->  “at different concentration of PVA aqueous solutions [125].

Author Response

Please find attached file with the detailed response to the Reviewer 2.

Round 2

Reviewer 1 Report

Dear Authors,

the manuscript has been meaningfully improved. Congratulations.

Kind regards.